

**Observed and projected impacts of coastal warming, acidification, and deoxygenation on Pacific oyster (*Crassostrea gigas*) farming: A case study in the Hinase Area, Okayama Prefecture and Shizugawa Bay, Miyagi Prefecture, Japan**

Masahiko Fujii[1,2], Ryuji Hamanoue[2], Lawrence Patrick Cases Bernardo[1], Tsuneo Ono[3], Akihiro Dazai[4], Shigeyuki Oomoto[5], Masahide Wakita[6], and Takehiro Tanaka[7]

[1]Faculty of Earth Environmental Science, Hokkaido University, Sapporo, 0600810, Japan
[2]Graduate School of Environmental Science, Hokkaido University, Sapporo, 0600810, Japan
[3]National Research Institute of Far Seas Fisheries, Fisheries Research Agency, Yokohama, 2368648, Japan
[4]Center for Sustainable Society, Minamisanriku, 9860775, Japan
[5]Eight-Japan Engineering Consultants Inc., Okayama, 7008617, Japan
[6]Mutsu Institute for Oceanography, Japan Agency for Marine-Earth Science and Technology, Aomori, 0350022, Japan
[7]NPO Satoumi Research Institute, Okayama, 7048194, Japan

*Correspondence to*: Masahiko Fujii (mfujii@ees.hokudai.ac.jp)





**Abstract.** Coastal warming, acidification, and deoxygenation are progressing, primarily due to the increase in anthropogenic $CO_2$. Coastal acidification has been reported to have effects that are expected to become more severe as acidification progresses, including inhibiting formation of the shells of calcifying organisms such as shellfish. However, compared to water temperature, an indicator of coastal warming, spatiotemporal variations in acidification and deoxygenation indicators such as pH, aragonite

saturation state ($\Omega_{arag}$), and dissolved oxygen in coastal areas of Japan have not been observed and projected. Moreover, many species of shellfish are important fisheries resources, including Pacific oyster (*Crassostrea gigas*). Therefore, there is concern regarding the future combined impacts of coastal warming, acidification, and deoxygenation on Pacific oyster farming, necessitating evaluation of current and future impacts to facilitate mitigation measures. We deployed continuous monitoring systems for coastal warming, acidification, and deoxygenation in the Hinase area of Okayama Prefecture and Shizugawa Bay

in Miyagi Prefecture, Japan. In Hinase, the $\Omega_{arag}$ value was often lower than the critical level of acidification for Pacific oyster larvae, although no impact of acidification on larvae was identified by microscopy examination. Oyster larvae are anticipated to be affected more seriously by the combined impacts of coastal warming and acidification, with lower pH and $\Omega_{arag}$ values and a prolonged spawning period, which may shorten the oyster shipping period and lower the quality of oysters. No significant future impact of surface-water deoxygenation on Pacific oysters was identified. To minimize the impacts of coastal warming

and acidification on Pacific oyster and related local industries, cutting $CO_2$ emissions is mandatory, but adaptation measures such as regulation of freshwater and organic matter inflow from rivers and changes in the form of oyster farming practiced locally might also be required.



# 1 Introduction

## 1.1 Coastal warming

Since the industrial revolution of the mid-18th century, anthropogenic carbon dioxide ($CO_2$) emissions have increased (Intergovernmental Panel on Climate Change (IPCC), 2021) as a result of activities such as fossil-fuel consumption, industry, and land-use changes (e.g. Le Quéré et al., 2018). The $CO_2$ emitted has a greenhouse effect and is therefore a contributor to global warming. Global warming is progressing due to the increase in anthropogenic $CO_2$ and other greenhouse gases. The IPCC 6[th] Assessment Report (AR6) stated that global surface temperatures were 1.09°C higher in 2011–2020 than in 1850–1900 (IPCC, 2021). In addition, ocean temperatures are increasing as the oceans absorb the increased thermal energy associated with global warming (e.g. Levitus et al., 2009). The rate of change is reported to be 0.56°C per century on a global scale and 1.28°C per century in the seas around Japan (Japan Meteorological Agency website; https://www.data.jma.go.jp/gmd/kaiyou/data/shindan/a_1/japan_warm/japan_warm.html). There is concern that the impact on ecosystems in the seas around Japan will be considerable.

The effects of rising sea temperatures on ecosystems vary. Most marine organisms are heterotherms, and there have been reports at higher latitudes of organisms that usually prefer warmer seawater in the south. For example, warm-water yellowtail landings have increased in Hokkaido, a northern island of Japan (Makino and Sakurai, 2012; Miyama et al., 2021; Fujii, 2022a) and corals are moving northward at a rate of 14 km/year in Japan's mainland (Yamano et al., 2011), indicating that the fishing and tourism industries are being affected by rising seawater temperatures.

There is also concern that this rise in temperatures may cause earlier or longer spawning and maturation times. For example, the Pacific oyster (*Crassostrea gigas*) reaches sexual maturity when the accumulated water temperature reaches 600°C based on a water temperature of 10°C, and that at water temperatures of 20°C or higher they spawn once and then mature and spawn again (Oizumi et al., 1971). There is concern that an earlier and longer spawning period may result in a mismatch with existing oyster-farming approaches.

Global warming may cause extreme events such as larger typhoons (e.g. Yoshino et al., 2015) and increased heavy rainfall (e.g. Papalexiou and Montanari, 2019). Increased high- rainfall events result in increased river flooding and alter material inputs to the ocean, thus affecting coastal ecosystems (Hoshiba et al., 2021), which may, in turn, affect human wellbeing via fisheries and marine tourism. Therefore, it is necessary to predict the impact of ocean warming on coastal areas and ecosystems, and to implement appropriate adaptation measures.

## 1.2 Ocean and coastal acidification

$CO_2$ leached into the ocean reacts with water ($H_2O$) in seawater to form carbonic acid ($H_2CO_3$). The $H_2CO_3$ separates into hydrogen ions ($H^+$), bicarbonate ions ($HCO_3^-$), and carbonate ions ($CO_3^{2-}$), releasing $H^+$ into seawater. Therefore, as the amount of $CO_2$ leached into the ocean increases, seawater, which is inherently slightly alkaline, decreases in pH and becomes closer to neutral or acidic. This phenomenon is called ocean acidification (Orr et al., 2005; Bates et al., 2014; Jiang et al., 2019).


Ocean acidification is a global phenomenon. Over the past century, global average pH values have decreased by 0.1 unit, indicating an increase in hydrogen ion concentrations ($[H^+]$) of nearly 30% (Orr et al., 2005; Doney et al., 2020). Additionally, rates of ocean acidification have been reported to vary by region. For example, in the Canary Islands in the North Atlantic Ocean, pH values reportedly decreased at a rate of $0.0017 \pm 0.0005$ per year from 1995 to 2003 (González-Dávila et al., 2007).

In coastal areas of Japan, pH values reportedly decreased by $0.0024 \pm 0.0042$ per year from 1978 to 2009 relative to the maximum observed pH (Ishizu et al., 2019). Thus, rates of acidification progression vary regionally, especially in Japanese coastal regions, compared to the North Atlantic. A major contributor to the differences in the progression of acidification in coastal areas is human activities such as coastal protection works, inflows of river water containing industrial wastewater, and sea-surface aquaculture (Suzuki et al., 2020). In addition, spatiotemporal variations in seawater pH are more pronounced in

coastal areas than in open-ocean areas because of the complex environments created by natural phenomena such as biological activity and river inflows associated with rainfall. Alterations in the acidity of coastal waters is termed coastal acidification or coastal ocean acidification (Wallace et al., 2014) and is typically distinguished from ocean acidification.

The spatiotemporal variations in coastal acidification are unclear; attempts have been made to clarify them by observing pH values and the calcium carbonate ($CaCO_3$) saturation state ($\Omega$), an indicator of acidification, using continuous fixed-point

monitoring in the Japanese coastal zone (e.g., Yamamoto-Kawai et al., 2015 and 2021; Christian and Ono, 2019; Wada et al., 2020; Fujii et al., 2021; Ishida et al., 2021; Wakita et al., 2021; Fujii, 2022b). These attempts have been hampered by the uneven distribution of sites and the short monitoring period, which precludes assessment of long-term trends due to climate change; further expansion of sites is needed.

The $H^+$ in seawater reacts with $CO_3^{2-}$ to maintain equilibrium. Therefore, the concentration of carbonate ions ($[CO_3^{2-}]$) in

seawater decreases as acidification progresses. Calcifying organisms such as shellfish, corals, shrimps, and crabs, which have shells and skeletons of $CaCO_3$, are affected by this process. Because calcifying organisms form their own shells and skeletons using calcium ions ($Ca^{2+}$) and $CO_3^{2-}$ in seawater, $\Omega$ values are an indicator of the effects on these organisms. Therefore, $\Omega$ and pH values are important for evaluating the effects of acidification on organisms. $\Omega$ is determined by the product of $[CO_3^{2-}]$ and calcium ion concentration ($[Ca^{2+}]$), which is expressed by the following equation:

$$\Omega = \frac{[Ca^{2+}][CO_3^{2-}]}{K_{sp}},\qquad(1)$$

where $K_{sp}$ is the solubility product of $CaCO_3$ (Guinotte and Fabry, 2008).

Calcifying organisms include commercially important species that provide significant ecosystem services, such as shellfish and corals. Therefore, there are concerns regarding the impact of acidification on human communities. In addition, $CaCO_3$ has two crystalline body structures, aragonite and calcite, with aragonite being the more soluble (Morse et al., 1980). Because the

larval stages of shellfish and corals form aragonite shells and skeletons, there is concern that the effects of acidification will be more pronounced than in organisms with calcite shells. Previous studies have reported the effects of reduced aragonite saturation ($\Omega_{arag}$) on different species, based on laboratory experiments that evaluated acidification effects such as coral bleaching and the occurrence of deformities and mortality in larval shellfish by manipulating the partial pressure of $CO_2$



(Anthony et al, 2008, Kurihara et al., 2007; Kurihara, 2008, Kimura et al., 2011; Onitsuka et al. 2014, 2018; Waldbusser et al.,
2015). However, it is not clear when and where these effects occur in the ocean. Therefore, to assess the acidification impact
on commercially important species, it is necessary to clarify the ocean environment and evaluate the impacts on each species
and life stage.

### 1.3 Deoxygenation

Climate change has increased the vertical density gradient of upper-ocean layers, thereby weakening the downward flux of
oxygen and hence decreasing the oxygen content. The decreased solubility of oxygen in seawater induced by ocean-surface
warming has contributed to the decrease in ocean oxygen content (ocean deoxygenation; Stramma et al., 2010, 2011, 2012,
2020; Helm et al., 2011; IPCC, 2019; Sasano et al., 2015, 2018; Ito et al., 2017; Schmidtko et al., 2017; Oschlies et al., 2018;
Ono et al., 2021). In coastal areas, by contrast, oxygen content is frequently disturbed by anthropogenic processes such as
eutrophication, changes in freshwater loading, and alternation of topography (coastal deoxygenation; Rabalais et al., 2010;
Zhang et al., 2010; Ning et al., 2011; Breitburg et al., 2018; IPCC 2019 Laffoley and Baxter, 2019; Wei et al., 2019; Limburg
et al., 2020; Xiong et al., 2020; Fujii et al., 2021; Kessouri et al., 2021). Climate change also affects the coastal oxygen
environment by increasing the temperature of coastal water, thus decreasing oxygen solubility. Changes in rainfall modulate
estuarine circulation in coastal areas, significantly affecting ecosystems and biogeochemical cycles (Guo et al., 2021; Hoshiba
et al., 2021). Climate change also modulates basin-scale water circulation, thereby changing the patterns and strengths of
seasonal intrusions of open-ocean waters into coastal areas (Koslow et al., 2011, 2015; Booth et al., 2012). These indirect
consequences of global climate change make coastal oxygen environments more problematic, even if the degree of
anthropogenic perturbations in coastal areas remains constant. Strict measures to diminish anthropogenic stresses to coastal
environments are thus required to maintain satisfactory oxygen conditions in coastal waters.

In Japan, nutrient loadings from land areas have gradually decreased in most coastal regions (Abo and Yamamoto, 2019).
Eutrophic conditions are however still extant in many bays and estuaries, and seasonal hypoxic conditions in summer bottom
layers improve only slowly (Imai et al., 2006; Ando et al., 2021; Yamamoto et al., 2021). Deoxygenation and ocean
acidification cause combined effects on marine organisms (Melzner et al., 2013; DePasquale et al., 2015; Gobler and Baumann,
2016; IPCC, 2018). Monitoring variations in oxygen and pH is thus essential for assessment of conditions in coastal ecosystems.

### 1.4 Oyster farming and the impacts of coastal warming, acidification, and deoxygenation

Oyster farming occupies an important position in the domestic fisheries industry in Japan. In 2018, the value of oyster
production from marine aquaculture was about JPY 35 billion, accounting for about 7% of Japan's total marine aquaculture
production. Furthermore, Hiroshima Prefecture, which ranked first in terms of production by prefecture, accounted for 73% of
total marine aquaculture production, and Okayama Prefecture, which ranked fourth, accounted for 35% (Ministry of
Agriculture, Forestry and Fisheries website). Thus, oyster farming is an important local industry, on which the region is highly
dependent. Therefore, there are concerns regarding the economic impacts of coastal warming, acidification, and deoxygenation



on regions where oyster farming is a key industry.

Previous assessments of the effects of acidification on Pacific oysters (*Crassostrea gigas*) have shown increased larval mortality and malformation rates due to lower pH and $\Omega_{arag}$ values, as well as reduced calcification rates in adult oysters (Gazeau et al., 2007; Kurihara et al., 2007; Waldbusser et al., 2015; Gimenez et al., 2018; Durland et al., 2019). Oyster farms

in northwestern Oregon, which generate USD 273 million annually, have been impacted by coastal upwelling causing deep, low-pH, low-$\Omega_{arag}$ seawater to manifest at the surface (Barton et al., 2012). In Hawaii, producers have responded by constructing onshore hatcheries and taking adaptive measures to avoid the effects of acidification on the vulnerable larvae stage by using artificial nursery practices (Barton et al., 2015). There is concern that Japan may face a similar situation in the future as acidification progresses, and there is a need to establish appropriate adaptation measures before the effects are felt.

Such adaptation measures include genetic modification, relocation of fish farms (Tan and Zheng, 2020), and reducing local dependence on industries that farm calcifying organisms (Ekstrom et al., 2015), although cost-benefit analyses are lacking.

### 1.5 Objectives

Although the ecological effects of coastal warming, acidification, and deoxygenation on Pacific oyster (*Crassostrea gigas*) are becoming clearer, when and how these effects will occur at oyster-farming sites are unknown. Because Pacific oyster is a

commercially important species, to recommend adaptation measures requires projection of future impacts of coastal warming, acidification, and deoxygenation. For this purpose, we used monitoring sites in Pacific-oyster-farming areas in Japan and developed a coupled physical-biogeochemical model (Chapter 2). Chapter 3 provides observed and projected data on coastal warming, acidification, and deoxygenation, and on Pacific oyster and farming thereof. Our findings are discussed and summarized in Chapters 4 and 5, respectively.






## 2 Materials and Methods

### 2.1 Study sites

Two sites of Pacific oyster (*Crassostrea gigas*) aquaculture were selected: the Hinase area in Bizen City, Okayama Prefecture (hereafter Hinase) and Shizugawa Bay in Minamisanriku Town, Miyagi Prefecture (hereafter Shizugawa) (Fig. 1). Okayama

and Miyagi Prefectures together account for approximately 20% of the total domestic oyster aquaculture production, making them important regions for domestic oyster aquaculture. Of these, Hinase accounts for 50% of Okayama Prefecture's oyster aquaculture production, and Shizugawa is a major oyster-farming area, accounting for 10% of Miyagi Prefecture's oyster aquaculture production (Ministry of Agriculture, Forestry and Fisheries website).

Hinase is located in the Seto Inland Sea, the largest enclosed coastal sea in Japan (The Association for Environmental

Conservation of the Seto Inland Sea website). The Seto Inland Sea is shallow, with an average depth of 38 m, and is bordered by open sea at its southeastern, northwestern, and southwestern ends. In addition to being a closed sea area, excessive inflow of nutrients from the land due to human activities since the 1950s, loss of seaweed and eelgrass due to land reclamation, and frequent red tides caused by these factors have led to eutrophication of the sea area, and hypoxia and anoxia in the bottom layer. Eutrophication has been overcome in many areas of the Seto Inland Sea by measures to control excessive inflow of

nutrients from the land, but exchange of seawater with the open sea is weak, and the bottom layer is hypoxic.

Shizugawa Bay is a medium-sized bay that measures approximately 10 km east to west and 5 km north to south, with a mouth facing east (Horii et al., 1994). The maximum water depth in the bay is 54 m (Ministry of the Environment, 2010; Komatsu et al., 2018). The bay is affected by the complex water movements caused by mixing of the Oyashio Current flowing from the north offshore of the bay, the Tsugaru Warm Current moving southward through the Tsugaru Straits, and the water

mass derived from the Kuroshio Current from the south. The tidal currents in Shizugawa Bay are slow, ranging from 5 to 15 cm s$^{-1}$ even at high tide, and the predominant currents flow in from the northern part of the bay and out of the southern part (Nagasawa et al., 1998). Wind- and density-driven currents drive active exchange of water from inside and outside the bay throughout the year (Takahashi et al., 2018). Based on these characteristics, Shizugawa Bay has been classified as both an enclosed coastal sea (Ministry of the Environment, 2010) and an open-type bay (Komatsu et al., 2018). Since the 1990s,

environmental impacts such as anoxia due to overcrowding of coho salmon and Pacific oysters have been observed (Nomura et al., 1996). Subsequently, the Great East Japan Earthquake of March 11, 2011, caused major damage to the social infrastructure surrounding the bay as well as the aquaculture facilities in the bay, and the subsequent tsunami affected the seagrass and seaweed beds and tidal flats that support the fisheries.

In both areas, local residents have taken the initiative to improve the marine environment and practice "*sato umi*", which

is defined as "a sea where productivity and biodiversity have increased due to the addition of human labor" (Yanagi, 2006). For example, in Hinase, there were approximately 590 ha of eelgrass beds in 1950, but by 1985 that area had drastically decreased to approximately 12 ha. Eelgrass seeding activities by local fishermen began in that year, and the eelgrass beds have now recovered to about 250 ha (Tanaka, 2017). This human labor has increased the numbers of species and populations of fish



and shellfish, and improved biodiversity. In addition, in Shizugawa, after the Great East Japan Earthquake, the number of
oyster rafts used for oyster aquaculture and the cultivation density decreased, improving the growth efficiency of the oysters
and enabling them to be shipped in 7–10 months, compared to 3 years before the earthquake, thus improving productivity
(Komatsu et al., 2018).

Oysters assimilate organic matter by filtering granular organic matter and expelling the unassimilated organic matter as
feces. Because the assimilation efficiency of oysters declines with age, the amount of organic matter contained in feces
increases with oyster culture duration. As the organic matter in feces settles to the seafloor, the rate of oxygen consumption for
decomposition of the feces increases, leading to deoxygenation. In other words, reducing the density of aquaculture decreases
the environmental impact on the seafloor, alleviating coastal acidification and deoxygenation. This increases productivity and
reduces the environmental impact (Komatsu et al., 2018).

Against this backdrop, marine-environment observations are being conducted in both areas, with active cooperation by
local fishermen, within the framework of the Nippon Foundation Ocean Acidification Adaptation Project (OAAP;
http://nippon.zaidan.info/dantai/0611718/dantai_info.htm), to assess acidification and to develop adaptation measures.

Four research sites have been set up in Hinase and Shizugawa (Fig. 1). In Hinase, Site H-1 is located at the mouth of the
Chikusa River, the largest river in the study site. Site H-2 is an oyster seedling site, located near the mouth of Katakami Bay.
Site H-3 is an eelgrass bed, located at the mouth of Genji Bay. Site H-4 is the farthest offshore, with water depths of 10.2–12.4
m. In Shizugawa, Site S-1 is at the mouth of the Hachiman River, the largest river in the area. Site S-2 is a seaweed-farming
site, and Site S-3 is a nursery for oysters. Site S-4 is the farthest offshore, and has water depths of 15.5–16.9 m.

### 2.2 Observation

We have measured hourly water-temperature, salinity, and pH values at a depth of 1 m at each site in Hinase since August 29,
2020 and in Shizugawa since September 4, 2020, using instruments capable of continuous measurement. Dissolved oxygen
(DO) has also been monitored continuously at a depth of 1–1.5 m at one site in Hinase (H-2) and one in Shizugawa (S-3). A
conductivity and temperature sensor (INFINITY-CTW ACTW-USB; JFE Advantech) was used to measure temperature and
salinity hourly, while DO was measured hourly using a RINKO W AROW-USB (JFE Advantech). To measure pH, glass-
electrode pH sensors (SPS-14; Kimoto Electric) were used. The sensors were removed every 1–3 months for cleaning,
including removal of attached organisms, data collection, battery replacement, and calibration. See Fujii et al. (2021) for details
of the experimental design.

Water samples were collected when the sensors were maintained, and chlorophyll, total alkalinity (TA), dissolved inorganic
carbon (DIC), nutrients (nitrate [$NO_3$], nitrite [$NO_2$], ammonium [$NH_4$], phosphate [$PO_4$], and silicate [$Si$]) were measured ($Si$
was not assessed at Shizugawa).

TA and DIC values were obtained using a total alkaline titration analyzer (ATT-05 by Kimoto Electronic) and a coulometer
(Model 3000A; Nippon ANS) (Wakita et al., 2017, 2021; Fujii et al., 2021). The values were calibrated against certified
reference material provided by Prof. A. G. Dickson (Scripps Institution of Oceanography, University of California San Diego)



and KANSO TECHNOS. The pH (total scale) values at the *in situ* temperatures were calculated from the carbonate dissociation constants in Lueker et al. (2000) and temperature, salinity, TA, and DIC using CO2SYS (Pierrot et al., 2006).

During continuous monitoring of pH, together with correction of the absolute value, it is necessary to correct for the drift of the observed value (Yamaka, 2019; Fujii et al., 2021). In this study, the pH value of a pH sensor at time t (pH(t)) was obtained using the following equation (Hamanoue, 2022):

$$pH(t) = pH_m(t) + [(pH_{sample}(t_i) - pH_m(t_i)) + \{pH_{sample}(t_e) - pH_m(dt_e)$$

$$- (pH_{sample}(t_i) - pH_m(t_i))\}] \times \frac{t - t_i}{t_e - t_i}, \tag{2}$$

where $pH_m(t)$ represents the measured value of pH at time t; $pH_{sample}(t_e)$ and $pH_{sample}(t_i)$ are the pH values at the end time ($t_e$)
and start time ($t_i$) of each deployment, respectively, obtained by the seawater sample and sensor; $pH_m(t_i)$ is the pH value measured by the sensor at time $t_i$; $pH_m(dt_e)$ is the minimal or average pH value measured by the sensor for 24 hours prior to $t_e$. pH increases during the day due to photosynthesis, and decreases during the night due to respiration of organisms. If algae or other organisms adhere to the glass-electrode portion of the sensor, the effect of photosynthesis during the day is amplified, and the pH value is overestimated. To minimize calibration uncertainty due to this effect, the lowest daily value was used for
$pH_m(dt_e)$ if an effect of photosynthesis was observed in the previous 24 hours, and the average value was used if not.

    $\Omega_{arag}$ can be calculated using one of the following values in addition to water temperature and salinity—pH, TA, DIC, and $CO_2$ concentration in seawater. Of these, the TA and DIC values were calculated by the above when seawater was sampled, but such sampling was conducted only once or twice per month. Therefore, because the TA of seawater is highly correlated with salinity (e.g. Yamamoto-Kawai et al., 2015), a regression equation was calculated from the salinity and TA values of the
seawater samples (Fig. 2). Hourly TA values were estimated from hourly salinity data obtained from continuous observations. Hourly values of $\Omega_{arag}$ were calculated using CO2SYS (Lewis et al., 1998), together with water-temperature and pH values obtained from continuous observations. The maximum error for this process of determining alkalinity from salinity is about 10 μmolkg$^{-1}$ and 0.02 for alkalinity and $\Omega_{arag}$, respectively.

    To examine the effects of precipitation and freshwater inflow from rivers on the spatiotemporal changes in acidification
indices, precipitation data from the sites nearest to Hinase (Mushiage, Oku Town, Setouchi City, Okayama Prefecture) and Shizugawa (Shizugawa, Minamisanriku Town, Miyagi Prefecture, respectively) (Japan Meteorological Agency website; https://www.data.jma.go.jp/obd/stats/etrn/index.php) were obtained. The precipitation data were compared directly with the spatiotemporal changes in salinity, pH, and $\Omega_{arag}$ to verify whether variations were due to precipitation or inflow from rivers.

### 2.3 Microscopic examination of oyster larvae

Like other calcifying organisms, Pacific oyster (*Crassostrea gigas*) is particularly vulnerable to acidification at the larval stage. By incubating Pacific oysters in a high-$CO_2$ tank, Kurihara et al. (2007) revealed that acidified water inhibited the growth of D-shaped veliger larvae. Thus, microscopic examination of D-shaped veliger larvae enables assessment of the impact of acidification on Pacific oyster.





Microscopy examination of D-shaped veliger larvae collected using 50–100-μm mesh plankton nets was carried out in
Hinase and Shizugawa during the spawning season. In Hinase, the examination was performed at Hinase Fisheries Association
from July 4 to August 31, 2020 (n = 370) and from June 21 to October 1, 2020 (n = 244), and at Oku Fisheries Association
from July 11 to September 9, 2020 (n = 292), and from July 2 to August 30, 2021 (n = 156). In Shizugawa, microscopy
examination was performed at Kesennuma Miyagi Prefectural Fisheries Experimental Station from July 27 to September 2,
2020 (n = 60) and July 26 to September 6 (n = 70).

**2.4 Modeling**

To reproduce the coastal environment in Hinase and Shizugawa and to project future conditions, the Regional Ocean Modeling
System (ROMS) was used. Of the versions of ROMS, we chose CROCO (ver. 1.1; Jullien et al., 2019), which can perform
high-resolution simulations and account for various interactions, including atmosphere, tides, and bathymetry. In addition,
CROCO enables coupling of ROMS with the Pelagic Interaction Scheme for Carbon and Ecosystem Studies (PISCES; Aumont
260 et al., 2003), a marine ecosystem model, enabling calculation of biogeochemical as well as physical processes (Bernardo et al.,
2021; Hamanoue, 2022). The model is therefore suitable for simulating complex coastal marine environments.

The prognostic variables for the physical processes of the model were water temperature and salinity, and those for the
biochemical processes were DO, TA, DIC, and nutrients ($NO_3$, $PO_4$, Si). pH and $\Omega_{arag}$ were calculated from the values of water
temperature, salinity, TA, and DIC obtained by the model using CO2SYS. The unavoidable biases in model results of
265 prognostic variables relative to observed values were corrected using the procedure adapted by Yara et al. (2011) and Fujii et
al. (2021).

The model domain was set to 133° 38' 06" to 135° 47' 67" E and 33° 93' 24" N to 34° 79' 81" in Hinase and 140° 86' 10"
E to 142° 86' 20" E and 37° 59' 47" to 39° 76' 47" N in Shizugawa. The horizontal resolution of the models was approximately
2 km. The vertical coordinate system was σ- coordinate and the number of layers was 32. Bathymetry was derived using the
15 arc-second General Bathymetric Chart of the Oceans (GEBCO) 2021 dataset (GEBCO website; Table 1). Simulations were
carried out for present and future (2090s) conditions. The simulation was carried out for a 1-year period from May to April
(2000 to 2001 for present and 2099 to 2100 for future) and the daily mean results at 1 m depth were used for analysis and
comparison with the observed results.

**2.4.1 Boundary conditions**

The boundary conditions for water temperature, salinity, current velocity, and water level were taken from the Future Ocean
Regional Projection (FORP)-JPN02 version 2 dataset (Nishikawa et al., 2021), which has a horizontal resolution of 2 km, the
highest resolution for Japan to date. For the future greenhouse gas emissions scenario, we used the MRI-CGCM3 climate
prediction model outputs developed at the Meteorological Research Institute (Tsujino et al., 2017) under the Representative
Concentration Pathways (RCP) 2.6 and 8.5 scenarios (van Vuuren et al., 2011) of the Coupled Model Intercomparison Project
phase 5 (CMIP5; Taylor et al., 2012).





Table 1 lists the boundary conditions used in this study. The present boundary conditions of DIC concentrations were estimated from water temperature and DO concentrations using the regression equation of Watanabe et al. (2020). Those for the 2090s, the RCP 2.6 and 8.5 scenarios from a climate model (model description and results of CMIP5-20c3m experiments (MIROC-ESM); Watanabe et al., 2011) were used. The partial pressure of $CO_2$ in the atmosphere ($pCO_{2air}$) for the 2090s was

set to 420 and 900 ppm for the RCP 2.6 and 8.5 scenarios, respectively.

For boundary conditions of the other prognostic variables, the present-replicate values were given for the 2090s. The World Ocean Atlas (WOA) 2009 (Garcia et al., 2010a, 2010b) was used for Shizugawa as boundary conditions for DO and nutrient ($NO_3$, $PO_4$, Si) concentrations. Because the WOA 2009 lacks data for the Seto Inland Sea, the boundary values for DO and nutrient concentrations in Hinase were obtained from the Ministry of the Environment website (https://water-

pub.env.go.jp/water-pub/mizu-site/mizu/kousui/dataMap.asp). The boundary conditions of TA values were estimated from temperature and DO concentrations using the regression equation of Watanabe et al. (2020).



# 3 Results

## 3.1 Observed results

### 3.1.1 Water temperature

Water temperatures showed significant seasonal variations at both sites. In Hinase, the highest water temperature during the observation period was 32.3°C at H-2 on August 8, 2021 (Fig. 3 (a)). The highest water temperatures at the other sites in Hinase were observed in August 2020, with a temperature difference of 1.2°C between sites. The lowest water temperatures were observed in the middle of January, 2021: 6.2°C at H-1, 3.9°C at H-2, 5.6°C at H-3, and 7.3°C at H-4.

In Shizugawa, the highest water temperature during the observation period was 28.7°C at S-2 on August 6, 2021, and the highest water temperatures at the other sites were observed on September 8, 2020 or August 6, 2021 (Fig. 3 (b)). The lowest water temperature of 6.5°C was observed at S-1 on February 9, 2021. The difference between sites was about 0.8°C and 0.7°C for the maximum and minimum water temperatures, respectively.

### 3.1.2 Salinity

Salinity varied between 30.5 and 31.5 at sites in Hinase and between 32 and 34 in Shizugawa (Fig. 3 (a), (b)). Although no significant differences were observed among the sites in Hinase, salinity was generally higher at H-4 than at the other three sites throughout the year. Salinity sometimes decreased locally after rainfall at all four sites, but the frequency and degree of salinity decrease differed at each site both in Hinase and Shizugawa. Lower monthly minimum values were often observed at sites closer to the coast (H-1, H-2, and S-1), which were affected more by riverine freshwater discharge.

### 3.1.3 DO

DO concentrations showed significant seasonal variation, generally being high in winter and low in summer at all sites in Hinase and Shizugawa (Fig. 4 (a), (b)). Although the DO concentrations were above the lower threshold of the optimal DO range for Pacific oyster growth (203-269 µmol kg$^{-1}$; Hochachka, 1980; Fisheries Agency, 2013) in Shizugawa, they were often below the optimal range in summer and autumn in Hinase.

### 3.1.4 TA

TA values estimated from continuous salinity observations using the above-mentioned regression equation (Fig. 2) matched those determined by water-sample analysis at each site (Fig. 5 (a), (b)). The estimates implied a significant decrease in TA values, associated with a localized decrease in salinity as a result of rainfall and subsequent enhanced riverine discharge, that could not be captured by once-or twice-monthly water-sample analysis.



### 3.1.5 DIC

DIC values determined by water-sample analysis showed clear seasonal variation, being generally high in winter and low in summer (Fig. 6 (a), (b)), likely a result of the higher solubility of atmospheric $CO_2$ at low temperatures and more vigorous primary production, respectively. The DIC estimated from water temperature, salinity, and pH (and TA via salinity) showed similar fluctuations to the corresponding TA. By contrast, the estimated DIC showed abrupt changes at all sites that were not captured by water-sample analysis. Most of the abrupt changes in estimated DIC were downward, and a significant decrease occurred at all four sites in Hinase on July 13, 2021, after a major rainfall event.

### 3.1.6 pH

pH values varied widely during the observation period at all sites in Hinase and Shizugawa, with a marked decrease after rainfall (Fig. 7 (a), (b)). The extent of the post-rainfall decline in pH differed among the sites. In Hinase, the lowest pH was in September 2021, and pH values were lower at H-1, H-2, and H-3 than at H-4, which was the farthest offshore. After rainfall in September 2021, the lowest pH values at H-1 and H-2 were 0.2 units lower than those at the other two sites. In Shizugawa, the lowest pH value of 7.8 occurred in July and August 2021, at S-1 and S-3 (in the estuary and offshore, respectively).

### 3.1.7 $\Omega_{arag}$

$\Omega_{arag}$ varied significantly during the observation period at all sites in Hinase and Shizugawa (Fig. 7 (a), (b)). The temporal variability varied from site to site, with greater decreases at sites closer to the coast. The threshold for an acidification effect on Pacific oysters (*Crassostrea gigas*) is a $\Omega_{arag}$ value of 1.5 (Waldbusser et al., 2015). Below that threshold, the development of Pacific oyster larvae is able to be affected, with slower growth and higher mortality.

$\Omega_{arag}$ values < 1.5 were often detected in Hinase, especially at H-1 and H-2, which were close to the river. Furthermore, during the spawning season of Pacific oysters from July to October (Oizumi et al., 1971), values fell below that threshold locally; the lowest $\Omega_{arag}$ of 0.8 was observed at H-2, which is used as a nursery for oysters, and values remained below the threshold for 2 weeks. In Shizugawa, the $\Omega_{arag}$ value was below the threshold only in August 2021 at S-3 for 4 hours, coinciding with the spawning season of Pacific oysters. However, no morphological abnormalities were observed in the larvae from Hinase and Shizugawa, implying no effect of acidification (Fig. 8).

### 3.2 Modeling results

### 3.2.1 Present reproduction

The modeled temperature and salinity reproduced the observed seasonal fluctuations in Hinase and Shizugawa (Fig. 9 (a), (b)). However, the model did not reproduce observed sudden decreases in salinity. This was likely due to insufficient input of freshwater from rainfall and riverine water into the model. The modeled seasonal fluctuation was around 1 month behind observations in Shizugawa. The model–data mismatch may be a result of the internal variability of the climate model (Yara et



al., 2011), especially for the Pacific Ocean, which provided the boundary conditions used in this study.

The modeled DO, TA, and DIC values reproduced the observed seasonal fluctuations in Hinase and Shizugawa (Figs. 10 (a), (b) and 11 (a), (b)). However, the model did not reproduce the short-term fluctuations in biogeochemical parameters. This was mainly because the temporal resolution of the model output is 1 day, insufficient to resolve significant daily fluctuations in biogeochemical processes predominantly caused by biological activities, i.e., photosynthesis by phytoplankton, seagrass,

and seaweeds in the day and respiration of marine creatures at night. Although the spatial resolution of the model (2 km) is one of the highest for downscaling climate-model outputs, it is insufficient to reproduce spatial differences in biogeochemical-parameter values among the four sites in Hinase and Shizugawa. Also, the model-data mismatch for TA and DIC values, especially the failure to reproduce sudden decreases, resulted from insufficient input of freshwater from rainfall and riverine water into the model.

The modeled pH and $\Omega_{arag}$ values reproduced those observed (Fig. 12 (a) and (b)). However, similar to the other biogeochemical parameters, the model had difficulty in simulating short-term fluctuations. Because the model's pH and $\Omega_{arag}$ values are calculated from modeled temperature, salinity, TA, and DIC values, uncertainties in the latter could magnify or cancel out those in the former.

### 3.2.2 Future projection

The projected results for physical and biogeochemical parameters in the 2090s differed markedly between Hinase and Shizugawa and RCP scenarios (RCP 2.6 vs. 8.5) (Figs. 13 and 14).

In Hinase, the projected rise in water temperature for the rest of this century was slight (Fig. 13 (a)), so DO concentrations will not change significantly (Fig. 13 (c)). Similarly, salinity will not change by the end of this century, leading to no significant change in TA (Fig. 13 (b), (d)). Therefore, the significant decrease in pH and $\Omega_{arag}$ values from the present to the 2090s,

especially with the RCP 8.5 scenario, is likely caused by the large increase in DIC resulting from the increased atmospheric $CO_2$ concentrations towards the end of the century (Fig. 13 (e)). The projected results show that larvae of Pacific oysters (*Crassostrea gigas*) may experience a critical $\Omega_{arag}$ value year-round, with the RCP 8.5 scenario (Fig. 13 (g)). This severe condition could be alleviated if anthropogenic $CO_2$ emissions are cut sufficiently in accordance with the Paris Agreement (RCP 2.6 scenario). The projected results also imply no severe impact of deoxygenation on the growth of Japanese oysters, neither now nor in the 2090s, at least at 1-m depth.

now nor in the 2090s, at least at 1-m depth.

In Shizugawa, water temperatures are predicted to rise by the 2090s (Fig. 14 (a)), substantially decreasing DO concentrations (Fig. 14 (c)). Although salinity and TA values will not change from the present to the 2090s with any RCP scenario (Fig. 14 (b), (d)), DIC will increase significantly (Fig. 14 (e)). Therefore, similar to Hinase, $\Omega_{arag}$ value is predicted to decrease markedly towards the 2090s (Fig. 14 (g)), mainly because of the increase in DIC values. In Shizugawa, no severe

conditions for Japanese oysters are predicted with regard to DO concentrations, but $\Omega_{arag}$ values will be below the threshold (< 1.5) except in summer, unless anthropogenic $CO_2$ is reduced sufficiently.



## 4 Discussion

### 4.1 Projected impacts of coastal warming, acidification and deoxygenation

In Hinase. based on Oizumi et al. (1971), Pacific oysters are estimated to have stopped spawning between October 24 and November 4, 2020, and between October 25 and November 7, 2021 and to have begun spawning between June 8 and 19 in 2021. In Shizugawa, spawning is estimated to have ended between October 8 and 10, 2020 and between October 16 and 18, 2021, and to have begun between July 19 and 24, 2021 (Table 2). The current end and start dates were calculated successfully by the model, i.e., on October 24 and June 10 in Hinase, and on October 8 and August 7 in Shizugawa, respectively (Fig. 15).

390          Because estimation of the timing of end and start dates is dependent on water temperature, the timing may be altered by future coastal warming. Our model results imply that in Hinase the start date will be earlier in the 2090s than at present, by 3 days with the RCP 2.6 scenario and by 7 days with the RCP 8.5 scenario, whereas the end date will not change (Table 2; Fig. 15). In Shizugawa, the end date will be 14 days later than at present in the 2090s with the RCP 2.6 scenario and 60 days later with the RCP 8.5 scenario. With the RCP 2.6 scenario, the start date is projected to be 14 days earlier in the 2090s than at

present. With the RCP 8.5 scenario, the water temperature is projected to be above 10°C year-round in the 2090s; therefore, we could not estimate the start date based on Oizumi et al. (1971).

         Coastal warming and acidification may have synergistic impacts on Pacific oyster larvae. As mentioned above, coastal warming will lengthen the spawning period, which is the life stage most vulnerable to acidification. Therefore, Pacific oyster larvae may suffer from acidification more seriously and over a longer period. Our model results imply that the number of days

on which $\Omega_{arag}$ values are below the threshold of acidification for Pacific oyster larvae (1.5) in Hinase will increase from 12 days at present to 24 days with the RCP 2.6 scenario and to 365 days with the RCP 8.5 scenario in the 2090s (Table 3; Fig. 15). With the RCP scenario, 145 of the 365 days are during the spawning period. In Shizugawa, the number of days on which $\Omega_{arag}$ values are below 1.5 will increase from 0 days to 216 days from the present to the 2090s with the RCP 8.5 scenario, while with the RCP 2.6 scenario the number of days in the 2090s will be similar to the present. The duration of severe conditions might

be 2 weeks longer, considering that 2–4 weeks are required for Pacific oyster larvae to settle after birth (*e.g.*, Chanley and Dinamani, 1980; Tachi et al., 2013). The prolonged spawning period may shorten the oyster shipping period and lower their quality (Akashige and Fushimi, 1992), potentially damaging the oyster-processing industry.

         Compared to the combined impacts of coastal warming and acidification, our model results indicate that the impact of deoxygenation on Pacific oysters will be less severe, at least in surface water. The model results reveal that the number of days

on which DO concentrations are below the optimal range for Pacific oyster growth (< 203 µmol kg⁻¹) will increase in Hinase from 13 days at present to 19 and 21 days in the 2090s with the RCP 2.6 and 8.5 scenarios, respectively, and 0 days in Shizugawa at present and in the 2090s (Table 3).

### 4.2 Alleviation of impacts on Pacific oyster farming

No impact of acidification on Pacific oysters has been reported to date. However, our findings indicate such impacts by the



415 end of this century, implying the need for alleviation measures.

### 4.2.1 Mitigation

Reducing anthropogenic $CO_2$ is the most important action globally. Our modeling results show that compared to the RCP 8.5 scenario, the impact of acidification will be alleviated significantly if we can limit future $CO_2$ emissions in accordance with the Paris Agreement (RCP 2.6 scenario). This action is also important to mitigate future extreme events, the intensity and

420 frequency of which will increase with global warming (e.g. Kimoto et al., 2005). To mitigate global warming and extreme events, ways should be found to avoid excessive discharge of freshwater and organic matter from rivers to coasts, both of which reduce coastal pH and $\Omega_{arag}$ values.

### 4.2.2 Adaptation

Despite any mitigation measures, coastal warming, acidification, and deoxygenation could last for decades. Therefore, it is

425 necessary to implement multiple adaptation measures in parallel.

At sites close to rivers, especially at H-2, extremely low observed pH and $\Omega_{arag}$ values were primarily caused by riverine freshwater inflow. Also, riverine organic matter dissolves at the coast, further lowering DO, pH, and $\Omega_{arag}$ values. In addition, extreme events are anticipated to occur more frequently and intensely in future, possibly increasing freshwater and organic matter inflow from rivers. Therefore, regulation of inflow to coasts is required, especially during the larval stage of Pacific

430 oysters, to alleviate acidification and deoxygenation at sites close to rivers. Restoration or placement of eelgrass beds or seaweed farms near river mouths and oyster seedling areas could trap river inflow of freshwater and suspended organic matter.

Our model results imply the need to adopt measures to change oyster-farming practices. Such measures include raising larvae, which are vulnerable to acidification, under good conditions that are maintained naturally or artificially. This is promising; some oyster-farming companies on the west coast of the United States, which experiences low pH and $\Omega_{arag}$ values

435 due to predominant coastal upwelling, have purchased oyster seed raised in Hawaii (higher pH and $\Omega_{arag}$ waters) (Barton et al., 2015).

Oyster farms are usually moved offshore before early autumn to stimulate oyster growth (Komiyama, 2002). However, this procedure is hampered by severe storms increasingly frequently. Extreme events such as severe storms are anticipated to occur more frequently and intensely in the future. Therefore, changes in oyster-farming practices may be necessitated by the

440 effects of climate change.



**5 Conclusion and Remarks**

This is the first study of the current and future impacts of coastal warming, acidification, and deoxygenation on Pacific oyster farming in Japan. In Hinase, oyster-farming sites have experienced critical levels of acidification, which have not affected

445   larvae. It may be necessary to revisit the acidification threshold for Pacific oysters farmed on the Japan coast.

Our future projections imply that unless $CO_2$ emissions are reduced in accordance with the Paris Agreement (RCP 2.6 scenario), oyster farming at the study sites may be deleteriously affected by coastal warming and acidification by the end of this century. The greatest impact will be on larvae, as a result of longer exposure to lower pH and $\Omega_{arag}$ waters. A prolonged spawning period may harm oyster processing by shortening the shipping period and reducing oyster quality. Therefore, to

450   minimize impacts on Pacific oyster farming, in addition to mitigation measures, local adaptation measures—such as regulation of freshwater and organic matter inflow from rivers and changes in oyster-farming practices—may be required.

Climate-change-driven extreme events will cause more frequent and intense heavy rainfall; subsequent river inflow of freshwater and organic matter to coasts may further reduce pH and $\Omega_{arag}$ in oyster farms. To plan how to minimize the adverse impacts of coastal warming and acidification, coupled physical-biogeochemical models with higher spatiotemporal resolution

455   are needed to simulate river-inflow processes and daily fluctuations in biogeochemical parameters.



**Author contributions**

TT launched the research project; AD and SO performed the measurements; MW analyzed the samples; LPCB and RH performed the modelling; MF, RH, and TO analyzed the data; RH and MF wrote the manuscript draft; LPCB, TO, AD, SO, 460 MW, and TT reviewed and edited the manuscript.

**Competing interests**

The authors declare that they have no conflict of interest.

**Acknowledgments.**

We thank Wakako Takeya for support. This study was supported by the Nippon Foundation Ocean Acidification Adaptation 465 Project (OAAP), the Integrated Research Program for Advancing Climate Models (TOUGOU; Grant Numbers JPMXD0717935498 and JPMXD0717935715), and the Advanced Studies of Climate Change Projection (SENTAN; Grant Number JPMXD0722678534), the Ministry of Education, Culture, Sports, Science, and Technology (MEXT) of Japan, and the Hokkaido University Functional Enhancement Project. This study used the Future Ocean Regional Projection dataset, which was produced by the Japan Agency for Marine-Science and Technology (JAMSTEC) under the SI-CAT project (Grant 470 Number JPMXD0715667163) of the Ministry of Education, Culture, Sports, Science and Technology of Japan. FORPJPN02 version 2 was provided by JAMSTEC and was collected and provided under the Data Integration and Analysis System (DIAS), which was developed and operated by a project supported by the Ministry of Education, Culture, Sports, Science and Technology, Japan. We used a coupled physical-biogeochemical model that was constructed under the framework of the Study of Biological Effects of Acidification and Hypoxia (BEACH) of the Environment Research and Technology Development 475 Fund (Grant Number JPMEERF20202007) of the Environmental Restoration and Conservation Agency of Japan.



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









**Table 1: Boundary conditions for the coupled physical-biogeochemical model used in this study.**

| Parameter | Dataset | Source |
|---|---|---|
| Bathymetry | The 15 arc-second General Bathymetric Chart of the Oceans (GEBCO) 2021 dataset | GEBCO website |
| Tide | TPXO Global Tidal Models (TPXO7.0) | Egbert and Erofeeva (2002) |
| Ocean physics (water temperature, salinity, current velocity, water level) | Future Ocean Regional Projection (FORP)-JPN02 | Tsujino et al. (2017) Nishikawa et al. (2021) |
| Atmospheric forcing (irradiation, air temperature, relative humidity, precipitation, wind velocity) | Hinase: GPV/JMA Meso-scale Spectral Model (MSM) | Japan Meteorological Agency website |
| | Shizugawa: Comprehensive Ocean-Atmosphere Data Set (COADS) 2005 | Da Silva et al. (1994) |
| Atmospheric $CO_2$ concentration | Present: 370 ppm Future: 420 ppm (RCP 2.6 scenario)          900 ppm (RCP 8.5 scenario) | van Vuuren et al. (2011) |
| Dissolved oxygen (DO) Nutrients ($NO_3$, $PO_4$, Si) | Hinase: Public water area water quality measurement data | Ministry of the Environment website |
| | Shizugawa: World Ocean Atlas 2009 | Garcia et al. (2010a, 2010b) |
| Total alkalinity (TA) | Present: obtained from the following equation: DIC = 2319 + 0.5155 T – 0.2367 DO where T: water temperature; DO: dissolved oxygen concentration Future: assume that the alkalinity does not change from present | Watanabe et al. (2020) |
| Dissolved inorganic carbon (DIC) | Present: obtained from the following equation: DIC = 2407 – 12.20 T – 0.7851 DO | Lewis et al. (1998) Watanabe et al. (2020) |
| | Future: outputs from Model description and results of CMIP5-20c3m experiments (MIROC-ESM) (2086-2095) | Watanabe et al. (2011) |





**Table 2: End and start dates of Pacific oyster (*Crassostrea gigas*) spawning in Hinase and Shizugawa, estimated from observed present and modeled present and future water temperatures and based on Oizumi et al. (1971).**


| | | | Hinase | | Shizugawa | |
|---|---|---|---|---|---|---|
| | | | **End date** | **Start date** | **End date** | **Start date** |
| **Observation** | | | October 24-November 4 (2020) | June 8-19 (2021) | October 8-10 (2020) | July 19-24 (2020) |
| | | | October 25-November 7 (2021) | | October 16-18 (2021) | |
| **Model** | **Present** | **Model (present)** | October 24 | June 10 | October 8 | August 7 |
| | **2090s** | **RCP 2.6** | October 23 | June 7 | October 24 | July 24 |
| | | **RCP 8.5** | October 24 | June 2 | December 2 | ? |





**Table 3: Simulated numbers of days when DO and $\Omega_{arag}$ values were below the lower bound of the optimal range (< 203 µmol kg⁻¹; Hochachka, 1980; Fisheries Agency, 2013) and the threshold of acidification ($\Omega_{arag}$ < 1.5; Waldbusser et al., 2015) for Pacific oyster larvae in Hinase and Shizugawa. Numbers in parantheses for the threshold of acidification denote the numbers of days of overlap with the Pacific oyster spawning period (except for the 2090s with the RCP 8.5 scenario in Shizugawa, because the spawning period could not be identified).**

| Threshold | | | Hinase # of days | Shizugawa # of days |
|---|---|---|---|---|
| DO < 203 ($\mu$mol kg$^{-1}$) | Present | | 13 | 0 |
| | 2090s | RCP 2.6 | 19 | 0 |
| | | RCP 8.5 | 21 | 0 |
| $\Omega_{arag}$ < 1.5 | Present | | 12 (0) | 0 (0) |
| | 2090s | RCP 2.6 | 24 (0) | 0 (0) |
| | | RCP 8.5 | 365 (145) | 216 (?) |



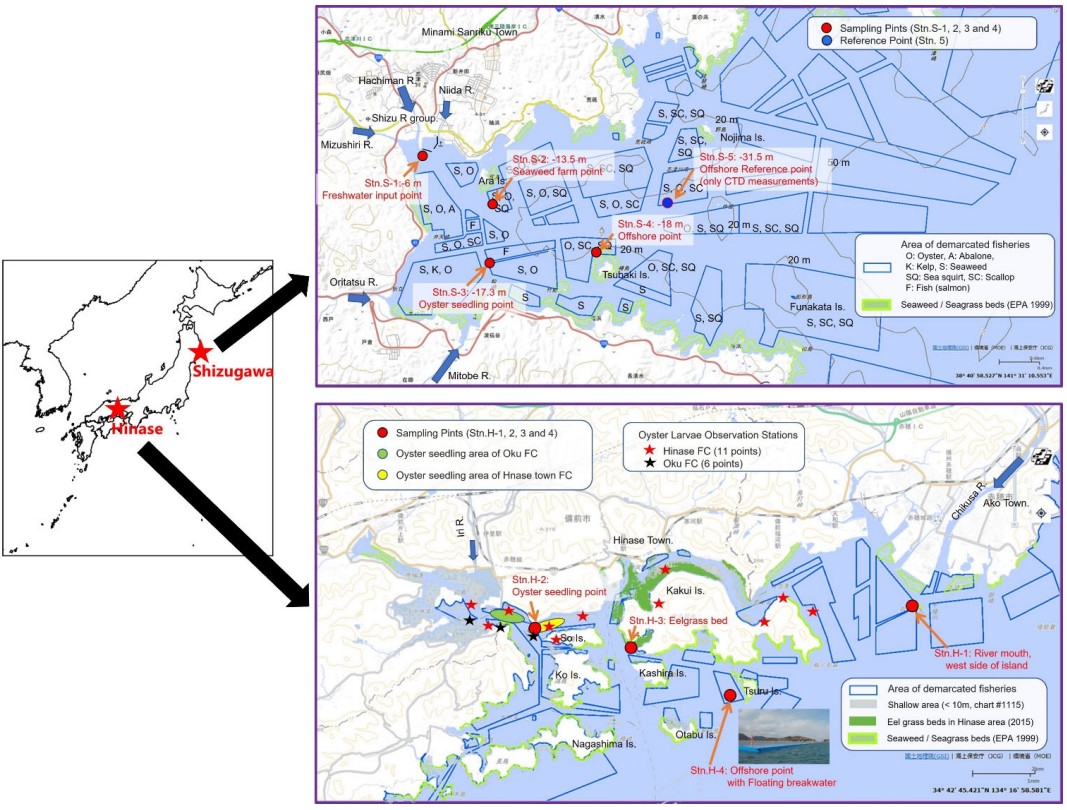


**Figure 1: Location of Hinase and Shizugawa Bay, and the H-1, H-2, H-3, and H-4 monitoring sites in Hinase and the S-1, S-2, S-3, and S-4 sites in Shizugawa Bay (maps created by the Nippon Foundation Ocean Acidification Adaptation Project (OAAP) based on MDA Situational Indication Linkages (https://www.msil.go.jp/)).**





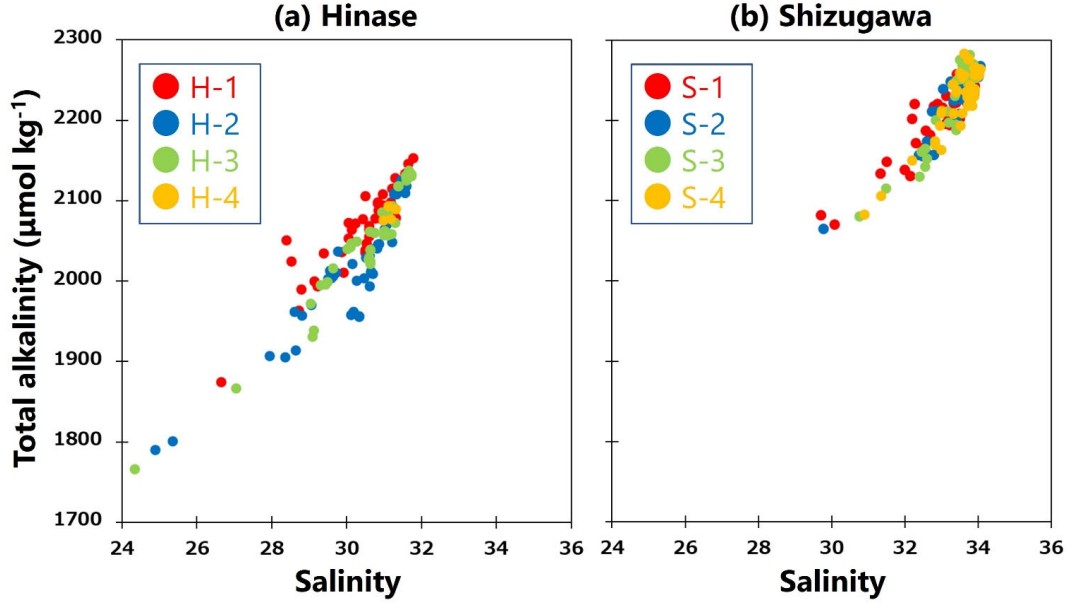

**Figure 2: Observed total alkalinity (TA) vs. salinity (dots) and regression lines and equations in (a) Hinase (H-1 [red], H-2 [blue], H-3 [green], and H-4 [orange]) and (b) Shizugawa (S-1 [red], S-2 [blue], S-3 [green], and S-4 [orange]). Correlation coefficients: H-1: $R^2 = 0.86$, H-2: $R^2 = 0.85$, H-3: $R^2 = 0.92$, H-4: $R^2 = 0.94$, S-1: $R^2 = 0.88$, S-2: $R^2 = 0.85$, S-3: $R^2 = 0.90$, and S-4: $R^2 = 0.90$.**



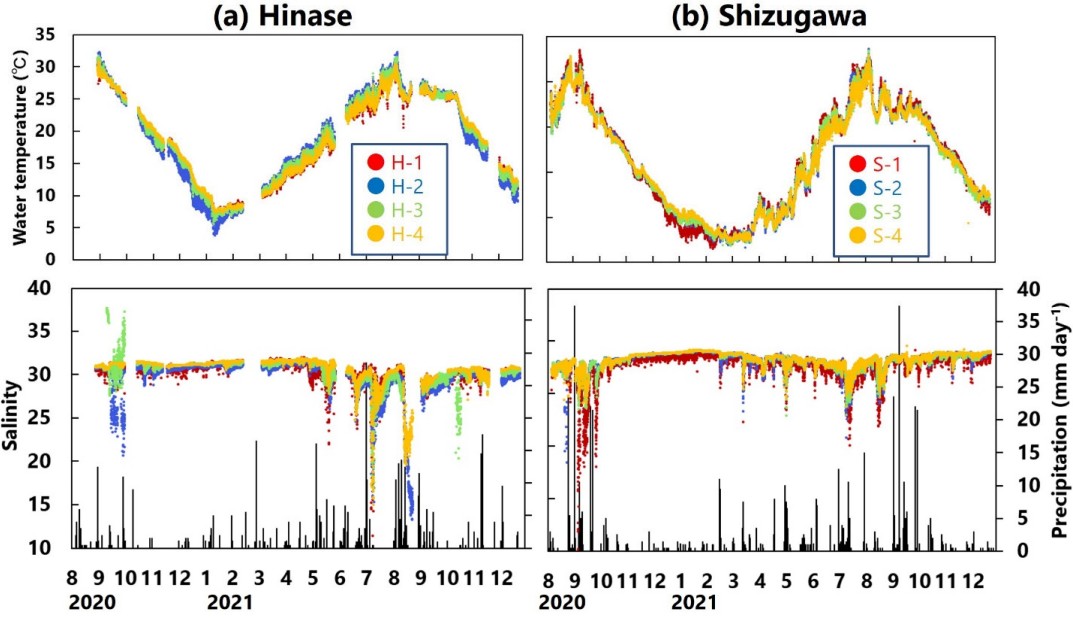

**Figure 3: Observed water-temperature (°C) (above) and salinity (below) values in (a) Hinase (H-1 [red], H-2 [blue], H-3 [green], and H-4 [orange]), and (b) Shizugawa (S-1 [red], S-2 [blue], S-3 [green], and S-4 [orange]) from August 2020 to December 2021. Black bars indicate daily precipitation (mm day⁻¹) at the nearest Automated Meteorological Data Acquisition System (AMeDAS) station—Mushiage (Hinase) and Shizugawa (Shizugawa) (Japan Meteorological Agency website; https://www.data.jma.go.jp/obd/stats/etrn/index.php).**





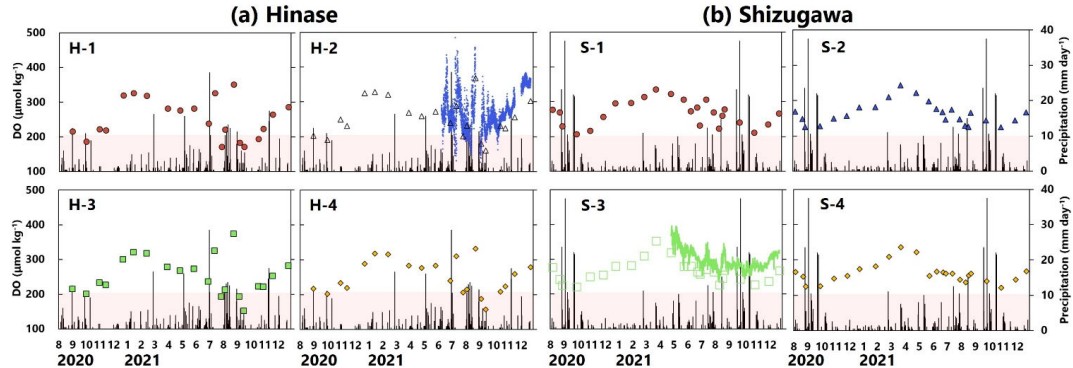


**Figure 4: Time series of dissolved-oxygen (DO; µmol kg$^{-1}$) values in Hinase and Shizugawa. Measurements were carried out when water-bottle samples were collected, and open circles (H-1 and S-1), triangles (at H-2 and S-2), squares (H-3 and S-3), and diamonds (H-4 and S-4) are measured values. Continuous monitoring using sensors was performed after**

**June 10, 2021 at H-2 and after April 27, 2021 at S-3. The monitored values are shown as dots (blue at H-2 and green at S-3). DO concentrations below the optimum DO range (203-269 µmol kg$^{-1}$) for the growth of Pacific oyster (*Crassostrea gigas*) (Hochachka, 1980; Fisheries Agency, 2013) are denoted in red. Black bars indicate daily precipitation (mm) at the nearest AMeDAS stations (Japan Meteorological Agency website; https://www.data.jma.go.jp/obd/stats/etrn/index.php).**






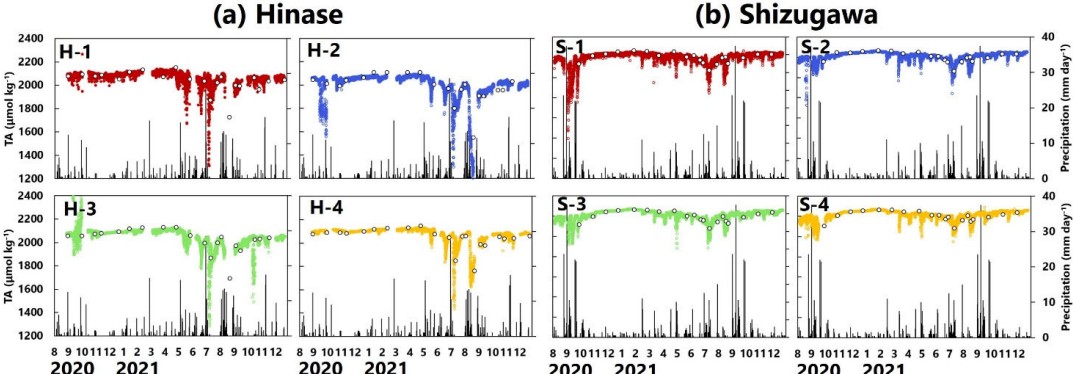

**Figure 5: Total alkalinity (TA) (µmol kg$^{-1}$) values based on water-sample analysis (open circles) and estimated from continuously observed salinity (colored dots) in Hinase (H-1 to H-4) and Shizugawa (S-1 to S-4) from August 2020 to**

**December 2021. Black bars indicate daily precipitation (mm) at the nearest AMeDAS stations (Japan Meteorological Agency website; https://www.data.jma.go.jp/obd/stats/etrn/index.php).**





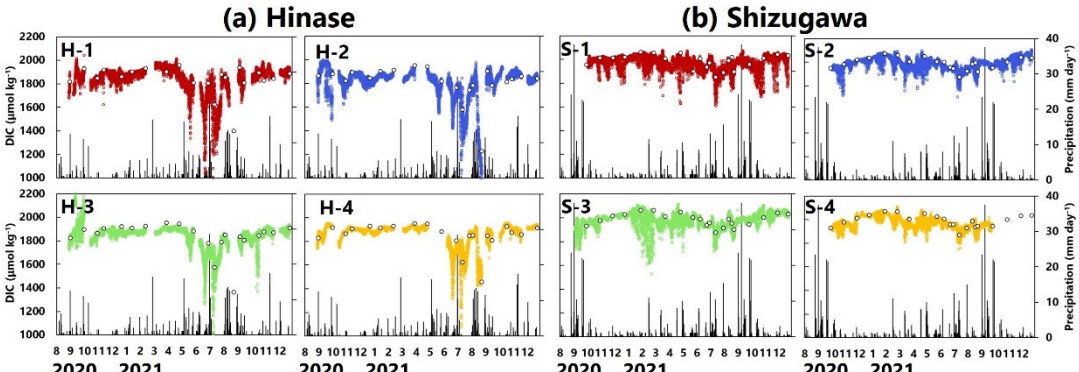

Figure 6: As Fig. 5, but for dissolved inorganic carbon (DIC) (µmol kg⁻¹) values.




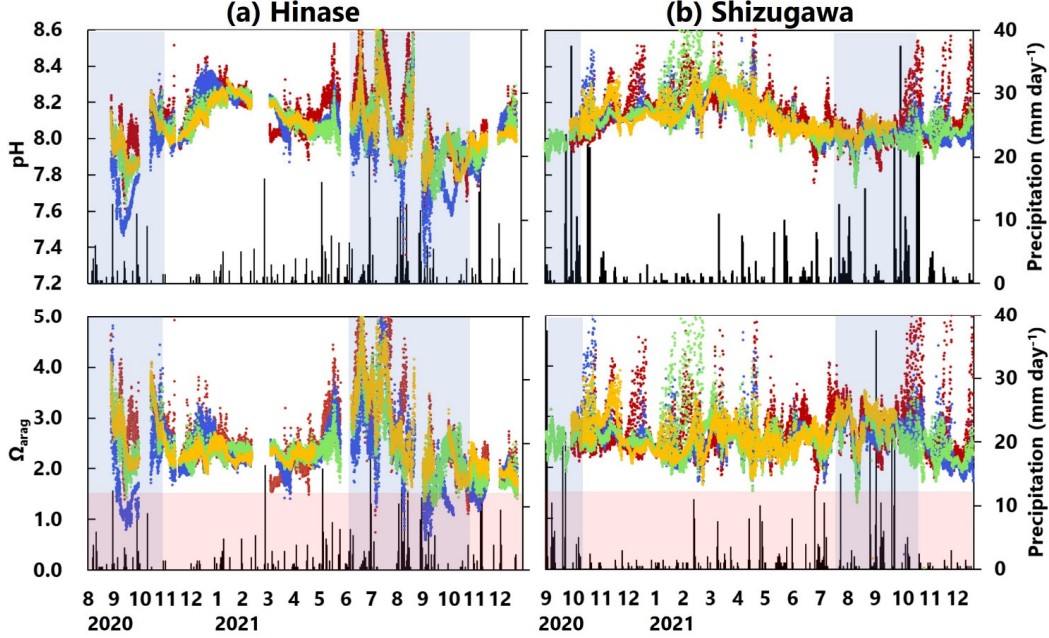

**Figure 7: Observed pH (top) and aragonite saturation state ($\Omega_{arag}$) (bottom) values in Hinase (left; H-1 [red], H-2 [blue], H-3 [green], and H-4 [orange]) and Shizugawa (right; S-1 [red], S-2 [blue], S-3 [green], and S-4 [orange]) from August**

**or September 2020 to December 2021. Red domains denote the critical level of acidification for Pacific oyster larvae in Waldbusser et al. (2015) ($\Omega_{arag}$ < 1.5). Blue domains denote the spawning season of Pacific oyster estimated from Oizumi et al. (1971). Black bars indicate daily precipitation (mm) at the nearest AMeDAS stations (Japan Meteorological Agency website; https://www.data.jma.go.jp/obd/stats/etrn/index.php).**






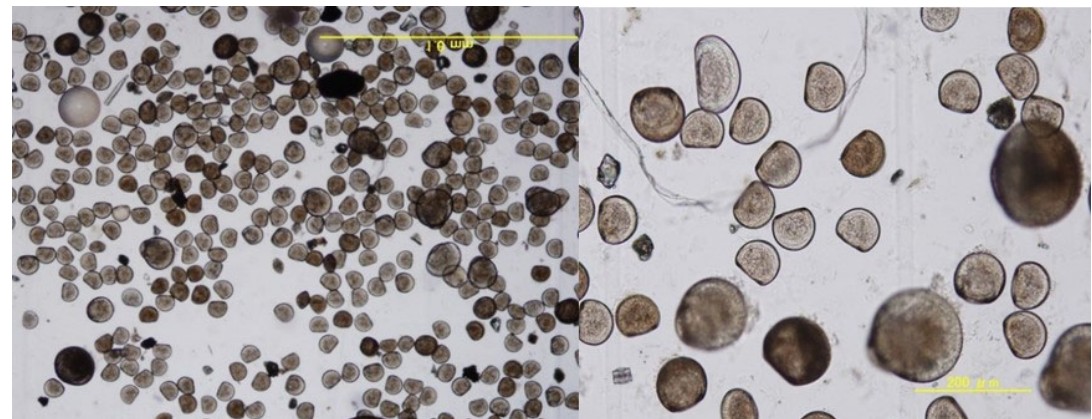

**Figure 8: Micrograph of Pacific oyster larvae in Hinase. No morphological abnormalities were observed.**

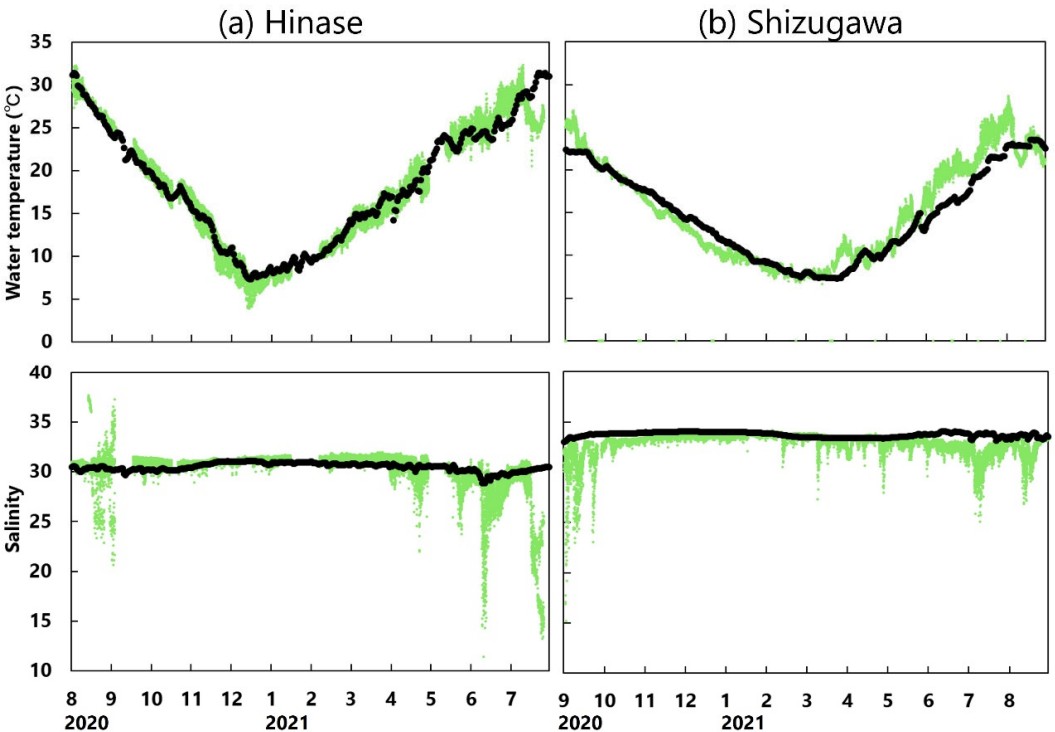


**Figure 9: Observed (green dots) and modeled (black lines) water-temperature (above) and salinity (below) values at 1-m depth in (a) Hinase (August 2020 to July 2021) and (b) Shizugawa (September 2020 to August 2021). The observed data at H-1, H-2, H-3, and H-4 in Hinase and at S-1, S-2, S-3, and S-4 in Shizugawa are combined in (a) and (b), respectively.**



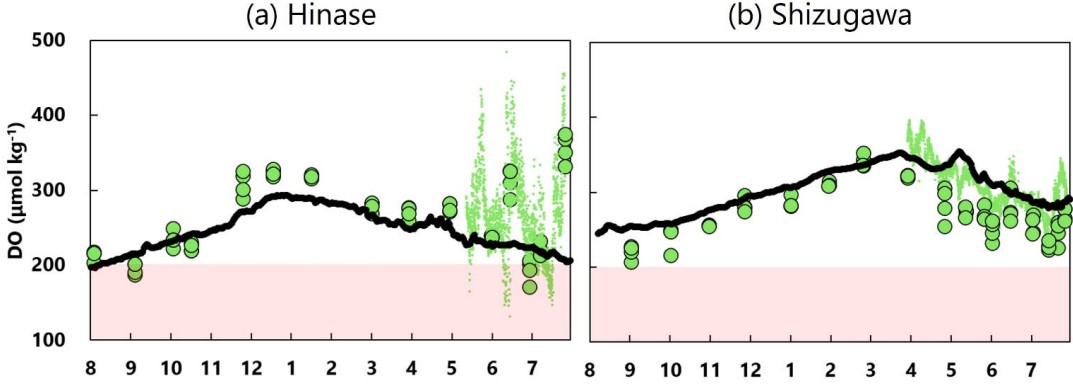


**Figure 10: Observed (circles and dots) and modeled (black lines) water temperature (above) and salinity (below) values at 1-m depth in (a) Hinase (August 2020 to July 2021) and (b) Shizugawa (September 2020 to August 2021). The measured and monitored values at H-1, H-2, H-3, and H-4 in Hinase and at S-1, S-2, S-3, and S-4 in Shizugawa are**

**combined and shown as green circles and dots in (a) and (b), respectively. DO concentrations below the optimum DO range (203–269 µmol kg⁻¹) for the growth of Pacific oyster (Hochachka, 1980; Fisheries Agency, 2013) are denoted in red domains.**



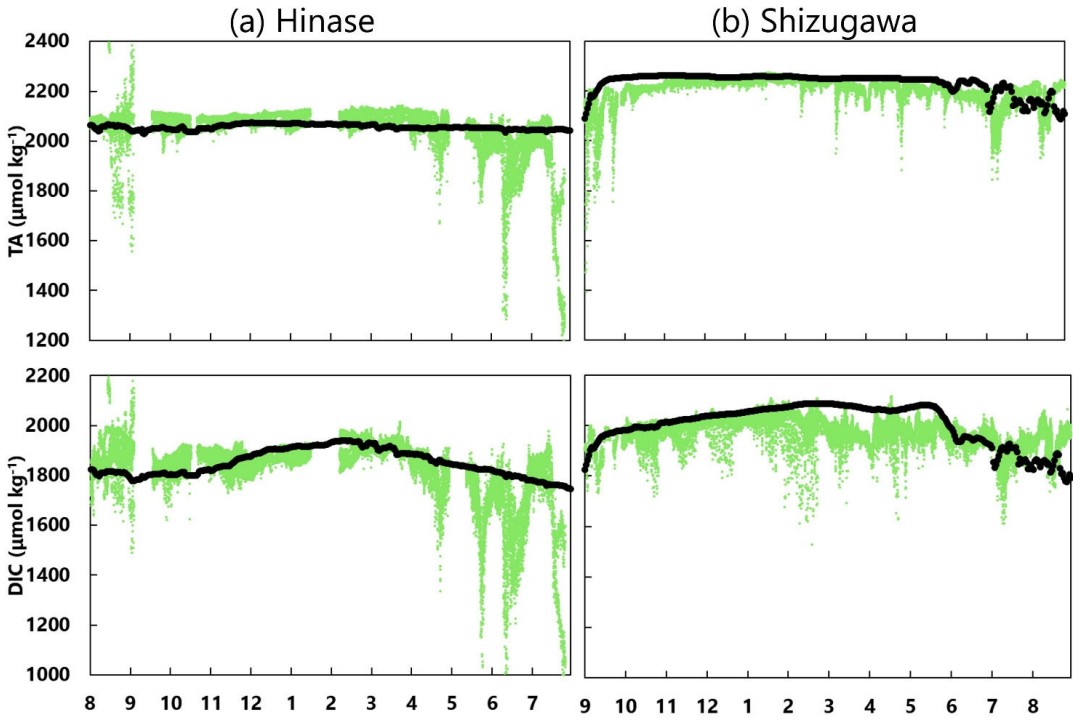

**Figure 11: As Fig. 9 but for TA (above) and DIC (below) values.**




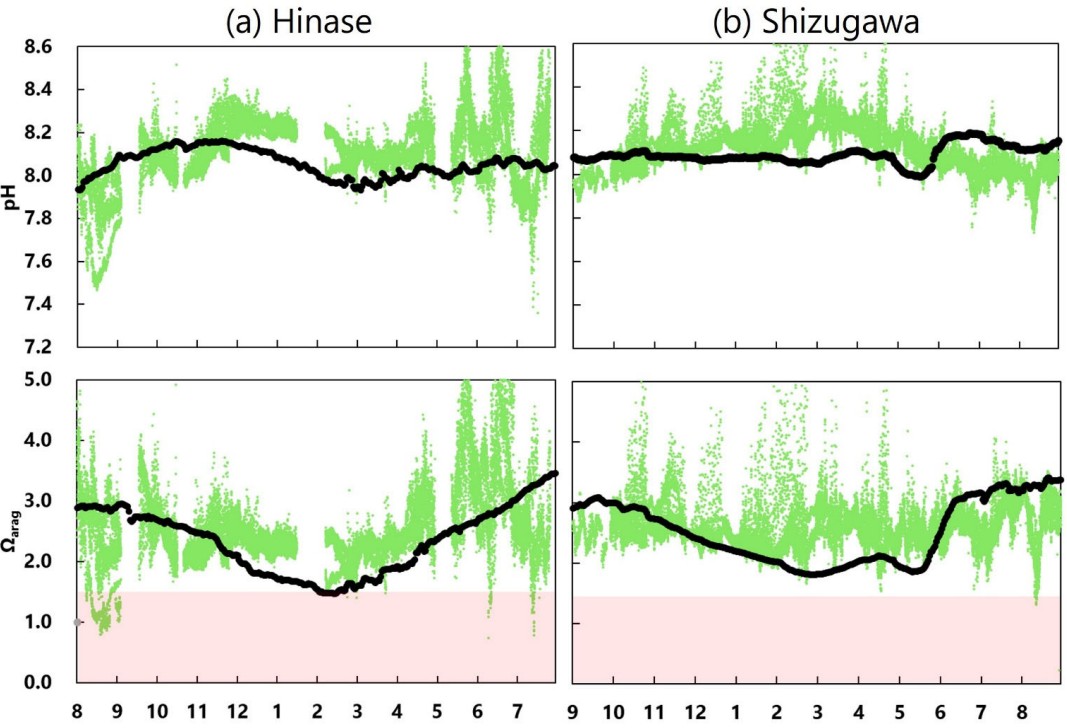

Figure 12: As Fig. 9 but for pH (above) and $\Omega_{arag}$ (below) values. Red domains denote the critical level of acidification for Pacific oyster larvae in Waldbusser et al. (2015) ($\Omega_{arag} < 1.5$).





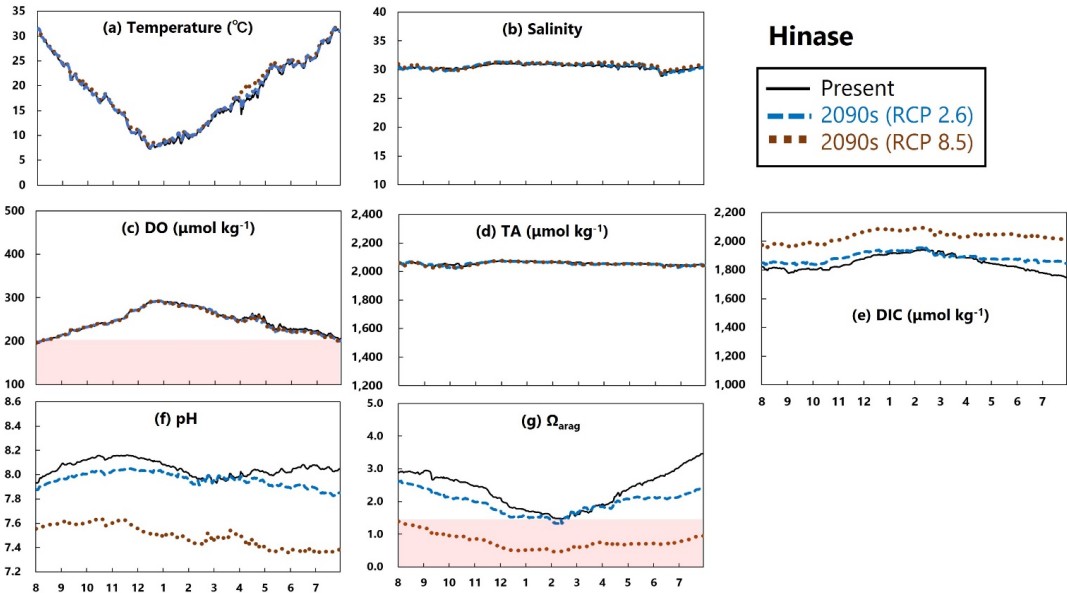

**Figure 13: Modeled (a) temperature (℃), (b) salinity, (c) DO (µmol kg⁻¹), (d) TA (µmol kg⁻¹), (e) DIC (µmol kg⁻¹), (f) pH, and (g) Ω_arag values in Hinase from August to July currently (black solid lines) and in the 2090s (RCP 2.6 scenario, blue dashed lines; RCP 8.5 scenario, brown dotted lines). Red domain in (c) denotes DO concentrations below the optimum DO range (203–269 µmol kg⁻¹) for the growth of Pacific oyster (Hochachka, 1980; Fisheries Agency, 2013). Red domain in (g) denotes the critical level of acidification for Pacific oyster larvae in Waldbusser et al. (2015) (Ω_arag < 1.5).**





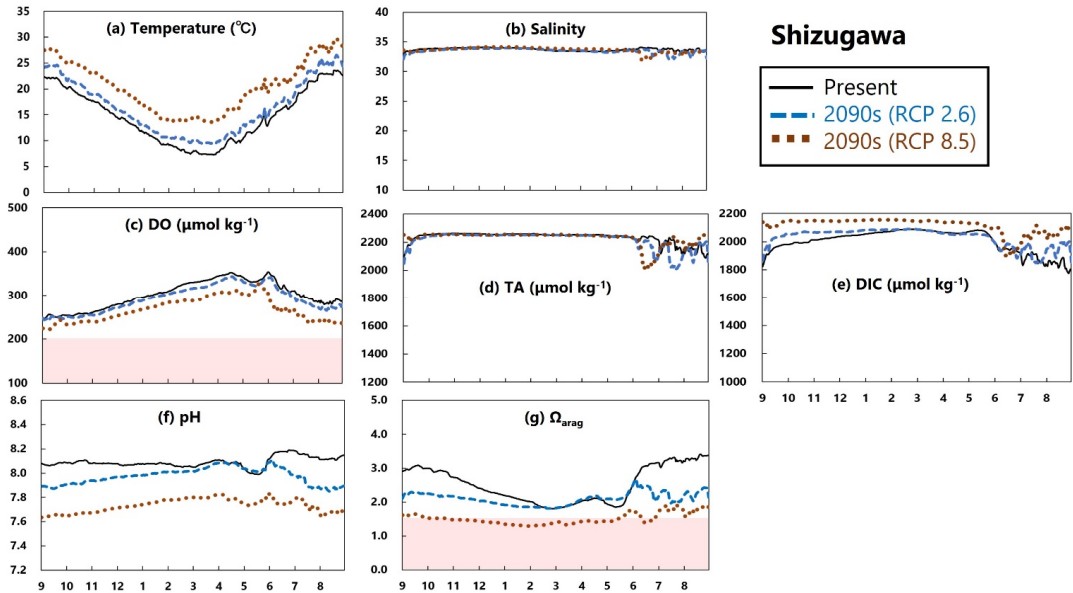

Figure 14: As Fig. 13, but in Shizugawa.





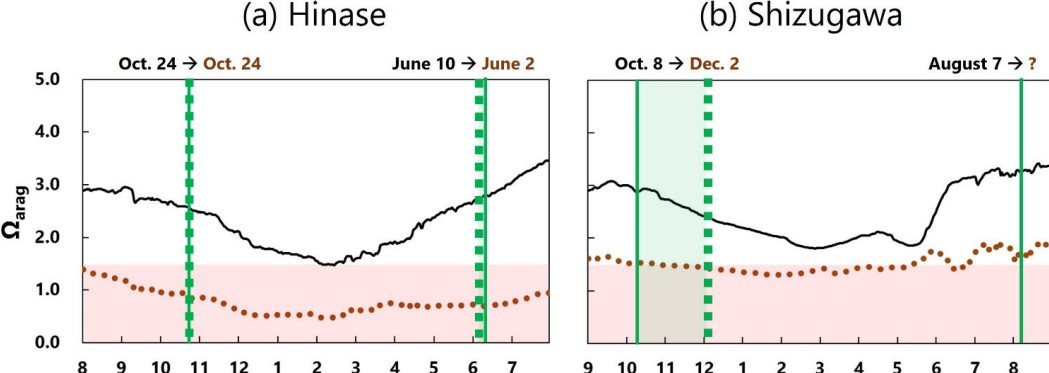


**Figure 15: Modeled $\Omega_{arag}$ values in (a) Hinase and (b) Shizugawa. Solid black and dotted brown lines denote those for the present and the 2090s (RCP 8.5 scenario), respectively. Red domains denote the critical level of acidification for Pacific oyster larvae in Waldbusser et al. (2015) ($\Omega_{arag} < 1.5$). Vertical green solid and dotted lines show the modeled start and end dates of the spawning season of Pacific oyster for the present and the 2090s, respectively, estimated from**

**Oizumi et al. (1971); green domains denote the projected prolonged duration of the spawning season from the present to the 2090s.**