# Peer review of "Assessing impacts of coastal warming, acidification, and deoxygenation on Pacific oyster (*Crassostrea gigas*) farming: A case study in the Hinase Area, Okayama Prefecture and Shizugawa Bay, Miyagi Prefecture, Japan"

_Biogeosciences, 2022_

## Author Response (AR1)

**Responses to Editor #1's comments**

First of all, thank you for all your comments that greatly helped us in improving our manuscript. Below are our responses. We have revised and re-submitted our manuscript based upon all the comments. Most of the previous figures have also been revised based on the following editor's comments we received just after submitting our previous manuscript:
* * *
*For "Figure 2" and "Figure 3": Please ensure that the colour schemes used in your maps and charts allow readers with colour vision deficiencies to correctly interpret your findings. Please check your figures using the Coblis – Color Blindness Simulator (https://www.color-blindness.com/coblis-color-blindness-simulator/) and revise the colour schemes accordingly.*
* * *
**Responses to general comments**

*Fujii et al. have aimed to characterise the chemical parameters of two coastal regions important for aquaculture in Japan. They have then used this data to model future scenarios, they then use existing published research on oysters to speculate how oysters and aquaculture may be affected.*

*Overall this is a valuable piece of work because such descriptions of costal habitats are lacking, yes these are the most productive marine aquaculture environments. This work would have been strengthened by experiments on oyster larvae replicating their modelled conditions, however, I understand that this would have included more work that might not have been feasible.*

*I raise this point because the manuscript currently relies on Waldbusser et al. to provide a critical Ωarg limit for larval survival, however, that work was completed in the USA, where local oyster genotypes are likely to differ significantly compared to those found in Japan. I think the levels of local adaptation to environments (especially when using wild oysters rather than selectively bred stock) should not be underestimated. There are many examples in the literature of physiological differences among genotypes of oysters, especially when sourced from different continents. I think the manuscript would benefit from more discussion regarding the biological relevance of the very comprehensive chemical measurements.*

→ (EC1-1) Thank you very much for all the informative comments. The authors have thoroughly discussed about the issue that our results regarding impacts of ocean acidification on Pacific oyster larvae might overestimate the reality in Japan coasts. As the editor pointed out, we used a threshold based on rearing experiments conducted in Oregon, USA, where Pacific oysters are not native and environmental conditions are considered to be different from Japan coasts. We have mentioned about this issue in Sections 2.5 (in Lines 242-246) and 4.3 (in Lines 398-402) in the revised manuscript).

**Responses to specific comments**

L306-307; Salinity results are stated to have "decreased" following rainfall. It would be good to describe here the extent that salinity decreased and for how long after significant rain. Furthermore, rainfall data is included in the figures. I suggest a statistical test to show the relationship between rainfall and salinity change, perhaps the two are not even correlated?

L309; how do you know that sites H-1, H-2 and S-1 were more affected by riverine flow? Please provide a statistical justification here.

→ (EC1-2, EC1-3) Based on the editor's and Reviewer #2's comments, the authors have rewritten the entire paragraph (Lines 266-280 in the revised manuscript). We tried to find statically significant relationships between salinity and rainfall, but could not find any. In particular, the difference in the salinity between the Hinase sites was not clearly elucidated, and we have removed the statement from the manuscript. While we still kept and have added some statements about the possibility of extremely low salinity caused by heavy rainfall by referring the timing of heavy rainfall and appearance of extremely low salinity at the Hinase sites and S-1 in the Shizugawa Bay, we made sure to mention that the relationship between the salinity and rainfall was not statistically significant.

L325; replace "downward"

→ (EC1-4) The authors have modified the phrase as: "*Abrupt drawdown of estimated DIC were sometimes found,*" (in Line 300 in the revised manuscript).

L336; as mentioned above, beware of placing too much emphasis on the Waldbusser results when your own study system and organisms are likely very different. I suggest addressing this in the discussion.

→ (EC1-5) The authors have set a section in the Methods part (*"2.5 Thresholds for evaluating the impacts on Pacific oysters (C. gigas)"*), and have moved all the statements about thresholds to this section. We have clarified that the Waldbusser et al. (2015)'s results used in this study were obtained from rearing experiments performed in Oregon, USA, of which Pacific oyster species and reactions to local environments may be different from those in Japan coasts as well (in Lines 243-245 in the revised manuscript).

L337; replace "able" with "likely"

→ (EC1-6) The word "able" has been replaced with "likely" (in Line 246 in the revised manuscript).

L347; this explanation belongs in the discussion not results.

→ (EC1-7) This explanation has been moved to Discussion section (4.3) as follows (Lines 407-409 in the revised manuscript): "*Also, considering that our current model underestimated observed sudden decreases in salinity as mentioned in 3.2, more realistic input data of freshwater from rainfall and riverine water would be necessary for better model performance.*"

L385; Considering that Oizumi et al., is not a widely available resource, the authors should include a description of how these estimates were made in the methods section, and then provide results on these estimates in the results section.

→ (EC1-8) Following the editor's comment, the authors have added a section in Methods part (*"2.5 Thresholds for evaluating the impacts on Pacific oysters (C. gigas)"*), and have explained how we applied the relation between the water temperature and spawning period of Pacific oysters obtained by Oizumi et al. (1971) in detail (Lines 236-241 in the revised manuscript). The relevant results have also been described in Results and Discussion sections (in Lines 261-265, 309-310, 323-328, and 365-372).

L400-406; This content and information belongs in the results section (not discussion) if it isn't already there.

→ (EC1-9) The authors have added the description of model results to Results section (in Lines 323-328 and 341-342 in the revised manuscript).

A figure here such as bar chart could be useful to display the mean increased/altered timing of reproduction among sites - this could replace one of the figures displaying measured parameters (these are less important in my opinion).
→ (EC1-10) The authors have replaced with a new figure (Fig. 16 in the revised manuscript).

L414; Reported by who? Please state if this is anecdotal evidence (i.e. oyster farmers) or your own observations.
→ (EC1-11) The statement was a bit exaggerated. The authors meant to say that there has been no anecdotal evidence of impacts of ocean acidification on Pacific oyster larvae found in Japan coasts to date. Therefore, we have revised the statement to: *"there has been no anecdotal evidence of impacts of ocean acidification on Pacific oyster larvae found in Japan coasts to date."* (in Lines 430-431 in the revised manuscript).

I also think that is not sufficient to use the absence of any morphological abnormalities (as found in your study) to report no effects on oysters. Significantly reduced larval supply could still have been a result, with abnormal larvae dying before your samples are taken. I think this limitation needs to be considered. Larval supply in oysters is a very difficult (almost impossible) thing to measure, however the use of settlement plates might have given a better indication of oyster recruitment.
→ (EC1-12) The authors felt that this is the weakest point of the study, so a major revision has been done to address this issue. We have developed further the explanation and discussion in the revised manuscript, especially in newly introduced Section 4.3 (*"Thresholds for impacts of ocean acidification on Pacific oysters in Japan coasts"*) to take into consideration the editor's helpful comments. In the two study sites, we did not use settlement plates but scallop shells to enhance oyster recruitment. We have also discussed about the possibility of larvae escaping from low salinity (and low $\Omega_{arag}$) waters (in Lines 403-407 in the revised manuscript). Although it does not seem that abnormal larvae existed as many were captured by plankton nets, we have also mentioned about the possibility of our failure to collect abnormal larvae samples for the reason that they died before our samples were taken (in Lines 394-397).

L420; There is some potential for local mitigation using plants etc. see Falkenberg et al., 2021
→ (EC1-13) Thank you for the comment as well as helpful references. The authors have added further description here about local mitigation of coastal acidification by using macroalgae and seagrasses with references (in Lines 436-438 in the revised manuscript).

Some of these options should be discussed here. I know that there is seagrass restoration occurring in the oyster aquaculture regions of Japan. Perhaps these measures may have some capacity for mitigation.
→ (EC1-14) Thank you for the suggestion. The authors have added description of *"For example, eelgrass restoration, that has long been performed in the Hinase Area as mentioned in 2.1, may have some capacity for mitigation."* in the end of this paragraph (in Lines 441-442 in the revised manuscript).

local catchment management is also another option - see Scanes et al., 2020.
→ (EC1-15) Following the comment, we have added the sentence *"Local catchment*

*management is also considered to alleviate the impacts of acidification and deoxygenation locally (e.g. Scanes et al., 2020)."* (in Lines 453-454 in the revised manuscript).

L424-440; Here there is no mention of the adaptive capacity of oysters themselves? Or the selective breeding work that is being undertaken to improve their capacity to withstand warming and acidification. I think that could also be raised here.

→ (EC1-16) In this manuscript, the adaptive capacity of oysters themselves was not mentioned because the authors do not have any evidence. Considering the editor's comment, we have added description about selective breeding work in the revised manuscript (in Lines 461-462).

L433; replace "good". This paragraph also has the opportunity to expand on the hatchery rearing of oysters and it's potential the alleviate issues with recruitment.

→ (EC1-17) The term "*good*" has been replaced with "*suitable*". The enhancement of hatchery rearing of Pacific oyster larvae has also been described in this sentence (in Lines 457-459 in the revised manuscript).

Figure legends – replace "pints" with "points" throughout.

→ (EC1-18) Corrected.

Figure legends – please use the full figure captions for all figures not "as figure xx".

→ (EC1-19) All the figure captions have been fully described in the revised manuscript.

Figure 1 – Eel grass is a type of seagrass but is labelled in addition to seagrass. Please clarify whether the "seagrass" labelled are also eelgrass or another species?

→ (EC1-20) The authors used the term "*seagrass*" several times in the previous manuscript. Most of them actually specifies "*eelgrass*", so the authors have replaced these with "*eelgrass*" in the revised manuscript.

References cited:
Scanes, E., Scanes, P., Ross, P.M., 2020. Climate change warms and acidifies Australian estuaries. Nature Communications 11(1), 1803
Falkenberg, L.J., Scanes, E., Ducker, J. and Ross, P.M., 2021. Biotic habitats as refugia under ocean acidification. Conservation Physiology, 9(1), p.coab077.

→ (EC1-21) The two references are informative for our study and have been cited in the revised manuscript. Thank you for the suggestion.

[Figure]

**Responses to Reviewer #1's comments**

First of all, thank you for all your comments that greatly helped us in improving our manuscript. Below are our responses. We have revised and re-submitted our manuscript based upon all the comments. Most of the previous figures have also been revised following your and the editor's comments.

**Responses to general comments**

*This manuscript by Fujii et al. submitted to Biogeoscience deals with current and future habitat for Pacific oyster (Crassostrea gigas) in two coastal sites in Japan. Recent reports that acidification has already negatively impacted on oyster growth along the West Coast of the United States are widely known among researchers. Also in Japan, the economic impact should be very large since oysters are a representative marine product. As such, the suggestion that the reduction of anthropogenic $CO_2$ can largely alter future habitat for oyster will be impressive for not only scientific community, but also for the general public.*

*Another commendable point of this paper is a successful long-term monitoring in coastal sites with several sensors. The figures presented in the manuscript suggest that the quality of data obtained by the sensors was good. As far as I know, there are not so much cases of such successful long-term monitoring in coastal sites. These observations should be maintained in the future.*

*My largest concern about this manuscript is the absence of long-term warming under RCP8.5 at Hinase. It is too unrealistic that future warming at Hinase is almost negligible (Figure 13a). There was no mention about future physical environment in the Seto Inland Sea in Nishikawa et al. [2021], which cited in the text. So, I could not verify whether negligible warming in this region is true or not. However, air temperature will likely increase over the long term under RCP8.5. It is difficult to believe that rising in air temperature will not affect water temperatures in the shallow Seto Inland Sea. I strongly urge the authors to check water temperature projections in the Seto Inland Sea.*

*Negligible warming in Hinase has resulted in much of the discussion being focused on whether or not there is an increase in water temperature. The differences in expected spawning period between Hinase and Shizuagawa appear to be due to the presence or absence of long-term warming. As this manuscript covered two cites, I want the authors to discuss the relationship between regional characteristics and expected changes in habitat for oysters. For example, Hinase is more enclosed area than Shizukawa. Do these differences in characteristics have any effect on acidification in the future? Unfortunately, there is little discussion about the impact of factors other than water temperature on environmental change at current manuscript.*

➔ (RC1-1) After receiving the reviewer's very helpful comments, we checked the boundary conditions for the future simulations for both Hinase and Shizugawa. And we found that we had not been able to account for future increases in air temperature for the Hinase simulations. Using air temperature data from FORP historical and future scenario runs, we were able to correct the previous atmospheric forcings we used. By simulating with the improved atmospheric conditions, the future projected results became more reasonable, with higher water temperature compared to the present (Figure 14 in the revised manuscript). However, it seems that the climate model outputs used as boundary conditions in our model

underestimate rise in water temperatures in the Seto Inland Sea. Therefore, the authors could not compare directly the projected environmental change, especially temperature and subsequent $\Omega_{arag}$ values between Hinase and Shizugawa.

As for the comparison of Pacific oysters between Hinase and Shizugawa, it is quite difficult to compare them directly, mainly because of the following two reasons: first, the oysters hatch and spawn eggs locally and have long been accustomed to the local environments in each site; second, relations between environmental characteristics and oysters' responses to environmental change have not yet been clarified well, except for those due to temperature, acidity and dissolved oxygen. Therefore, we focused only on discussing the difference between Hinase and Shizugawa in terms of the impacts of these parameters on oyster farming in our previous manuscript. However, referring the editor's comment, although the impacts of many factors on Pacific oysters are still unclear, the authors have added further discussion about the impacts of low salinity on Pacific oyster larvae (in Lines 403-409).

*Also, I think long-term oligotorophication in the Sato Inland Sea is a hot topic. I recommend that the authors mention regarding oligotorophication. If it is impossible, the author should mention as limitation of the projection in the text.*

➔ (RC1-2) The authors have mentioned regarding oligotrophication in Lines 127-129 in the revised manuscript. We have also added explanation about observed nutrients (in Lines 281-287 in the revised manuscript). However, there are no thresholds of nutrient concentrations to express oligotrophication, so instead, we referred to the half-saturation constants of nutrients used in the model, and suggested that $NO_3$ and $PO_4$ are considered to be depleted, which is regarded as an oligotrophic condition in some seasons in the surface water in both sites.

*I agree with the posted comment (by Dr. Ishizu) that discussion is too little in the current manuscript. The authors have competent observational data and computational methods. Further discussions utilizing these resources are needed for the acceptance.*

➔ (RC1-3) We have developed the Discussion section following your and Dr. Ishizu's comments in the revised manuscript. Firstly, future projection results have been discussed in Section 4.1 (in Lines 345-362 in the revised manuscript). Anticipated change in Pacific oyster's spawning period in the future has been described as a projected combined impact of coastal warming and acidification on Pacific oysters (in Lines 365-372). Also, following editor's comments, the authors have added a new section (*"4.3 Thresholds for impacts of ocean acidification on Pacific oysters in Japan coasts"*) and have discussed about thresholds used in this study to express the impacts of coastal warming, acidification and deoxygenation and their limitations (in Lines 390-409).

**Responses to specific comments**

*(Line 164) The reference about oligotrophication (i.e., an overcome of eutrophication) is needed. I think that Abo and Yamamoto [2019], which was already cited in the text is suitable.*

➔ (RC1-4) Based on the comment, we have referred to Abo and Yamamoto (2019) and Yamamoto et al. (2021) here, and have added a relevant statement (*"Eutrophication has been overcome in many surface waters of the Seto Inland Sea through measures to control excessive inflow of nutrients from land over the last few decades, and the surface waters are even oligotrophic nowadays (e.g. Abo and Yamamoto, 2019; Yamamoto et al., 2021), but exchange of seawater with the open sea is weak, and the bottom layer is hypoxic."*) in Lines 127-129 in the revised manuscript.

*(Line 207) Information about the calibration of DO sensor is needed. I suppose that it was done by two-point calibration at 0% and 100%.*

➔ (RC1-5) As the reviewer points out, the calibration of DO sensor was done by two-point calibration at 0% and 100%. We have added the following sentence to the Method section of the revised manuscript (in Lines 159-160): *"Calibration of the DO sensor was carried out by two-point (zero and span) calibration using 0 and 100% (saturated) oxygen waters (Fujii et al., 2021)."*

*(Line 237) "The maximum error ~ is about 10 μmol kg-1" Is this true? In Figure 2, some TA data appear to be deviated by more than 10 μmol kg-1 from possible regression line.*

➔ (RC1-6) Following the reviewer's comment, we have checked the values again, and as the reviewer pointed out they were found to be underestimated. Therefore, we have revised the maximum error of TA and $\Omega_{arag}$ to 30 μmol/kg and 0.06, respectively (in Line 191). Thank you for the helpful comment.

*(Line 305) "Although no significant differences were observed among the sites in Hinase, salinity was generally higher at H-4 than at the other three sites throughout the year." What do these sentences mean? Was the difference in salinity statistically insignificant? The author should clarify whether this difference is important in this study or not.*

➔ (RC1-7) All the statements in this paragraph were not clear and complete. Therefore, the authors have rewritten the entire paragraph (Lines 266-280 in the revised manuscript). Especially, we have paid an attention to mention that extremely low salinity seems to be related to heavy rainfall at some sites, but the relationship between the salinity and rainfall was not statistically significant.

*(Line 313) Does the upper limit of optimal DO range (269 μmol kg⁻¹) have any biological meaning? Most of observed DO exceeded this value (Fig. 4).*

➔ (RC1-8) As the upper limit is much less important than the lower limit with regard to biological implications and might also cause confusion, we have removed the upper limit from this sentence (in Lines 247-248), Figure 4 and the caption in the revised manuscript.

*(Line 341) "In Shizugawa, the $\Omega$arag value was below the threshold ~. However, no morphological abnormalities were observed ~." Then, what does the threshold mean?*

➔ (RC1-9) The two facts contradict each other. To address the issue, a major revision has been done. As mentioned above, the authors have further developed the explanation and discussion in the revised manuscript, especially in 4.3 (*"Thresholds for impacts of ocean acidification on Pacific oysters in Japan coasts"*). We have also discussed about the possibility of larvae escaping from low salinity (and low $\Omega_{arag}$) waters (in Lines 403-409 in the revised manuscript). Although it does not seem that abnormal larvae existed as many larvae were captured by plankton nets, we have also mentioned the possibility of our failure to collect abnormal larvae samples for the reason that they died before our samples were taken (in Lines 391-397).

*(Line 438) "Extreme events such as severe storms are anticipated to occur more frequently and intensely in the future." References are essential.*

➔ (RC1-10) The authors have added a reference (e.g. Kimoto et al., 2005; IPCC, 2022) (in Line 439 in the revised manuscript).

[Figure]

**Responses to Reviewer #2's comments**

First of all, thank you for all your comments that greatly helped us in improving our manuscript. Below are our responses. We have revised and re-submitted our manuscript based upon all the comments. Most of the previous figures have also been revised following your and the editor's comments.

**Responses to major comments**

*Reading through this manuscript, I thought that the volumes of each section are unbalanced. Especially, the volume for the result section is too short, compared to the volumes of introductions and methods.*

→ (RC2-1) The authors have revised the manuscript based on the comment, especially by adding further description to results section, including horizontal distribution of modeling results (as mentioned below) (in Lines 317-319 in the revised manuscript) and have also developed the discussion, including oligotrophication (following the other reviewer's comment; in Lines 281-287), observed low salinity (in Lines 266-280) and Pacific oyster spawning period (in Lines 261-265 and 323-328). On the other hand, the introduction and methods sections have been shortened appropriately, as mentioned below.

*I believe that one of the highlights in this study is to develop the model for the specific coastal area. Therefore it would be better to add more analysis by using the model outputs to show how their model is reproduced well in this target coastal area. At that time, seasonal horizontal distributions of each variable could be useful with comparison of the other observational data. Probably DIC and ALK are probably difficult to be got horizontally, but temperature and salinity, oxygen data could be available if you try to find in Japan. These improvements could make you deepen for your understanding of the model, which will make useful for your future study.*

→ (RC2-2) Following the comments, the authors have added figures and descriptions of horizontal distribution of model results for temperature, salinity and dissolved oxygen (in Lines 317-319 and Figs. 8 and 9 in the revised manuscript). Unfortunately, we could not obtain sufficient observational data in our study sites. Therefore, the authors performed comparison of model results with observational data, both of which were obtained in our study (in Figs. 10-13 in the revised manuscript).

**Responses to minor comments**

*Abstract: The sentence of the abstract should be blushed up. The current abstract was not a self-contained summary of your work. Method, how to examine and what you found should be included. The sentence "Coastal warming, acidification, ... to facilitate mitigation measures" can be shorten. The sentence "To minimize… oyster farming practiced locally might also be required" is not necessary.*

→ (RC2-3) Following the reviewer's comment, the authors have revised the entire abstract. The last sentence has been removed following your comment. The sentence "*Coastal warming, acidification, ... to facilitate mitigation measures.*" has been shortened as: "*Moreover, there is concern regarding the combined impacts of coastal warming, acidification, and deoxygenation on Pacific oysters.*" (in Lines 21-22 in the revised manuscript)

*Title: After revising the manuscript, I suggest you reconsider the title. The current title does not reflect the content of the current manuscript.*

→ (RC2-4) The authors have revised the title to: "*Assessing impacts of coastal warming, acidification, and deoxygenation on Pacific oyster (Crassostrea gigas) farming: A case study in the Hinase Area, Okayama Prefecture and Shizugawa Bay, Miyagi Prefecture, Japan*".

1. *Introduction: The sentences are too long. The sentence can be shorter. The current manuscript is divided into 5 sections (1.1, ~ 1.5), but I think it would be better to write it all in one. When you improve the manuscript, you also think about the balance of the volumes in the section. The volume of the introduction is heavy compared to the results, discussion and conclusion sections.*

→ (RC2-5) Based on the comment, the authors have combined all the sections together. Also, the section has been shortened by cutting descriptions that are too detailed and will be repeated in following sections.

*Study sites: This section is also too heavy, compared to the results, discussion and conclusion sections.*

→ (RC2-6) This section has also been shortened by cutting out overly detailed descriptions and summarizing the information about model boundary conditions using Table 1.

*Observed results: The current version has been divided the section into 3.1.1~3.l.7, but you don't need to divide individually. Please reconsider this part.*

→ (RC2-7) The authors have combined all the sections together.

*Modeling results: The results of the numerical models are necessary. Please put additional analysis such as horizontal distributions and so on to show how the model reproduces in this target area horizontally and timely.*

→ (RC2-8) Thank you for the helpful comments. Based on the reviewer's comments, the authors have added horizontal distributions of modeled temperature, salinity and dissolved oxygen (in Lines 317-319 and Figs. 8 and 9 in the revised manuscript). These results and comparison of time series of modeling results of individual parameters with the observed (in Figs. 10-13 in the revised manuscript) have been developed in Section 3.2 (*"Modeling results"*) in the revised manuscript.

*Future projection: This section is a discussion matter. This part can be moved to the discussion section if you add analyses of the model reproducibility.*

→ (RC2-9) This section has been moved to discussion section (Section 4.1) in the revised manuscript.

*Section 4.2: The section 4.2, 4.2.1 and 4.2.2 can be moved to the conclusion section. Please reconsider your construction.*

→ (RC2-10) Those sections have been moved to Conclusion part (*"5.1 Alleviation of impacts on Pacific oyster farming"*), following the reviewer's comment.

*Please add references after the sentence (Extreme events such as severe…intensely in the future) in page 16.*

→ (RC2-11) The authors have added references (e.g. Kimoto et al., 2005; IPCC, 2022) (in Line 439 in the revised manuscript).

*5. Conclusion: The conclusion should be improved. Basic conclusion includes the purpose, the summary of this study and self-evaluation/prospects.*
→ (RC2-12) The authors have revised the conclusion part ("*5. Conclusion and Remarks*"), mainly following the reviewer's valuable comments. Suggestions on mitigation and adaptation measures based on our study have mentioned in this part (*"5.1 Alleviation of impacts on Pacific oyster farming"*).

*Fig 1: The information of latitude and longitudes are necessary. Right figures are not appropriate in scientific papers. I think that making an original map by yourself is necessary. In that case, Japanese character should not be included in your map. Right figure is relatively too small.*

→ (RC2-13) Based on the comment, maps in Fig. 1 have been regenerated to more clearly show the most relevant information.

---

## Referee Report (RR1)

Although the manuscript has been updated better, I still have a major comment for this manuscript. I will therefore suggest the major revision to your editor.

**Major comments**

1) The understanding of the seasonal variability in pH and $\Omega$ is important in this. I therefore recommend authors to calculate the seasonal pH and $\Omega$ sensitivities for T, S and DIC and ALK. The following reference are helpful for your revision. They use the Taylor expansions of pH and $\Omega$ derivatives and evaluate the T, S, DIC and Alk dependence of pH and $\Omega$ values.

DeJong et al. (2015):
Equation (2) of the following article is about the talyor expansion of omega
https://bg.copernicus.org/articles/12/6881/2015/bg-12-6881-2015.pdf
Hagens and Middleburg (2016):
Equation (2) of the following article is about the talyor expansion of pH
https://agupubs.onlinelibrary.wiley.com/doi/full/10.1002/2016GL071719

This calculation makes you discuss your results in detail qualitatively and the science quality for this paper improve much better.

**Minor comments**

**Figures and Tables**

1) You need subsequent numbers (a, b, c..) in Fig. 1, 3, 4,5, 6, 7, 8, 9, 10, 11, 12 and 13 and add the subsequence figure number after your appropriate sentences, which is helpful for the readers to identify which figure we should see correctly.

2) The sizes of the title and scale are too small in Figs. 3, 4, 5, 6, 8, 9, 10,11, 12, and 13.

3) Figure 1: The topographic map is too rough. You can use the model topographic data. In addition, it is better to add the sea topography, the name of the river location and bays, you mentioned in the manuscript (in Section 2.1). If you cannot use the model topographic data, JOCD website provides the 500m-meshed topographic data (https://www.jodc.go.jp/jodcweb/index_j.html).

4) There are too many figures of time-series. Some of them are repeated in this manuscript. For example, Fig. 3, 4, 5 and 6 show the same black bars indicating hourly precipitation. I recommend you combine some of them into one figure, especially for the observation figures. And if you do so, you can make the bigger figures, which becomes reader's friendly.

5) There are no x-axis titles in Fig. 3, 4, 5, 6, 10, 11, 12, 13, 14, 15.

6) The resolution of Figure 7 is low. It is difficult to identify the yellow character inside the figure.

7) Table 1 : Please add the html addresses for GEBCO, Japan Meteorological Agency website, Ministry of the environment website, respectively.

8) Table 3 : Are the days when omega in Hinase and Shizugawa in the simulation with RCP8.5 opposite (365 and 216)? Please check it.

10) The caption about DIC has been forgotten in Figure 5.

**Chapter1**

1) In Chapter1, author explains the mechanisms how global warming occurs and ocean uptake carbon in the second and third paragraphs. I recommend you add the equation such as ($CO_2$ + $H_2O$ -> $H_2CO_3$ etc) after the appropriate sentences (2nd and 4th paragraphs).

2) 4th paragraph: Is the word "$CaCO_3$ saturation state ($\Omega$) values" is general word?

3) 5th and 10th paragraph: "① However, it is not clear when and where these effects occur in the ocean. Therefore to assess the acidification impact on commercially ······ and evaluate the impacts on each species" "② Although the ecological effects of coastal warming, acidification···..clearer, when and how these effects will occur at oyster-farming sites are unknown." These are duplicated. I think you can delete one of them (①).

**Section 2.1**

1) The 4th paragraph can be deleted, judging from the contents and balance of the manuscript.

2) 5th paragraph: If you revise the figure, I recommend you to add the location name "Chikusa River", "Katakami Bay", "Genju bay", and "Hachiman River", respectively.

**Section 2.4**

1) 2nd paragraph: The model domain area can be shown in Figure 1 with observational sites.

2) 3rd paragraph: Please add number in km for "15 arc-second".

**Section 2.4.1**

1) Section 2.4.1 can be combined into Section 2.4, which results in deleting Section 2.4.1.

**Section 2.5 and Table 2, 3, and 16**

I couldn't understand how the end date of the spawning season of pacific oysters with RCP 8.5 senario is determined, judging from your sentence in Section 2.5 "Pacific oysters reach

sexual maturity when the accumulated water temperature reaches 600 degree based on a water temperature of 10 degree, and that at water temperatures of 20 degree or higher they spawn ocean and then mature and spawn again". Is it possible to find the end of the spawning day, although we cannot find the start day with RCP 8.5 senario in Sizugawa?

**Section 3.1**
1) Please correct typo "oin" in the 1st paragraph in Section 3.1.
2) 2nd paragraph can be combined into the 1st paragraph.
3) Please add figure numbers after your sentence, which makes more readers friendly.
4) 4th paragraphs: you write "a statistically significante relation between the rainfall and salinity was not idfentified" and "the relation between the salinity and rainfall was not statistically significant at any of the sites in Hinase and Shizugawa, and future studies are necessary". I think that the reason why you cannot find the significant relationship is because you compared the total records when you compared. Please try to analyze with some ingenuities. For example, comparison of the short-term data, individually. When we see the time-series of salinity and precession visually, we can see that they seemingly have some relationships.
5) 5th paragraph: Is there no figure we should see? If so, please add the sentence "not shown".

**Section 3.2**
1) 1st paragraph: Please add some description about the horizontal distributions more.

**Section 4.2**
2nd sentence "Our model results imply that the number of ⋯ in Hinase will increase from .. to 365 days with the RCP8.5 scenario in the 2090s". Is this sentence correct? Please check it.

**Section 5.1 and 5.2**
These sections are written about the mitigation and adaption. I think that it may not include them in the scientific paper. I recommend you delete these sections.

---

## Referee Report (RR2)

I found the minor revision points, but after that, I think that it is time to accept this manuscript.

**Minor comments**

P9, l252-l253, I suggest you this revision below.

"In Hinase, Pacific oysters were estimated to have stopped spawning between October 24 and November 4, 2020, and between October 25 and November 7, 2021 and to have begun spawning between June 8 and19 in 2021, judging from the water temperature thresholds based on Oizumi et al. (1971)".?

P9, l252-253, Please add the Figure number we should see.

P12 eq.(4) please check the location of the comma(,).

P12, l345-l359, Fig14, please write which location you used when you draw Fig. 14.

P12, l361-l362, I wonder if you need the consideration from the effect of air-sea exchange as well as biological production?

---

## Author Response (AR2)

**Responses to Reviewer #1's comments**

Thank you for checking our previous manuscript very carefully and in detail, and giving very helpful and practical comments that greatly helped us in further improving our manuscript. We have revised and re-submitted our manuscript based upon all your comments. Most of the previous figures have also been revised following your and a second reviewer's comments. The information about an English certificate in the previous manuscript has also been deleted in the revised manuscript.

**Responses to major comments**

*1-1) The understanding of the seasonal variability in pH and $\Omega$ is important in this. I therefore recommend authors to calculate the seasonal pH and $\Omega$ sensitivities for T, S and DIC and ALK. The following reference are helpful for your revision. They use the Taylor expansions of pH and $\Omega$ derivatives and evaluate the T, S, DIC and Alk dependence of pH and $\Omega$ values.*

*DeJong et al. (2015): Equation (2) of the following article is about the talyor expansion of omega https://bg.copernicus.org/articles/12/6881/2015/bg-12-6881-2015.pdf*

*Hagens and Middleburg (2016): Equation (2) of the following article is about the talyor expansion of pH*

*https://agupubs.onlinelibrary.wiley.com/doi/full/10.1002/2016GL071719*

*This calculation makes you discuss your results in detail qualitatively and the science quality for this paper improve much better.*

➔ Thank you very much for this very useful comment. The authors have calculated monthly-mean contributions to pH and $\Omega_{arag}$ changes by temperature, TA, DIC and salinity, based on our model results, because monitoring results were often missing temporally. The analyzed results contain some interesting insights of similarities and differences between contributors and regions, as newly described in Section 3.2 (in Lines 345-373 of the revised manuscript).

**Responses to minor comments**

**Figures and Tables**

*1-2) You need subsequent numbers (a, b, c..) in Fig. 1, 3, 4,5, 6, 7, 8, 9, 10, 11, 12 and 13 and add the subsequence figure number after your appropriate sentences, which is helpful for the readers to identify which figure we should see correctly.*

➔ In Fig. 7, no numbering has been placed because one of the figures in Fig. 7 in the previous manuscript has been cut in the revised manuscript. The other figures have been modified following the reviewer's comment.

*1-3) The sizes of the title and scale are too small in Figs. 3, 4, 5, 6, 8, 9, 10,11, 12, and 13.*

➔ These figures have been enlarged following the reviewer's comment.

*1-4) Figure 1: The topographic map is too rough. You can use the model topographic data. In addition, it is better to add the sea topography, the name of the river location and bays, you mentioned in the manuscript (in Section 2.1). If you cannot use the model topographic data, JOCD website provides the 500m-meshed topographic data (https://www.jodc.go.jp/jodcweb/index_j.html).*

➔ The name of rivers and bays and the locations have been added to Fig. 1 in the revised manuscript. The topographic map has now been drawn using the GEBCO bathymetry dataset used to generate model bathymetry in this study (https://gebco.net/data_and_products/gridded_bathymetry_data/) which has data at 15 arc-second intervals (~500 meters near the equator).

➔

*1-5) There are too many figures of time-series. Some of them are repeated in this manuscript. For example, Fig. 3, 4, 5 and 6 show the same black bars indicating hourly precipitation. I recommend you combine some of them into one figure, especially for the observation figures. And if you do so, you can make the bigger figures, which becomes reader's friendly.*

➔ Considering the reviewer's comment, the authors have combined eight figures in Figs. 4 and 11 into two figures. On the other hand, although the 16 figures in Figs. 3, 5, 6, 10, 12, 13 consisted of four figures originally (when we first submitted the manuscript last year), we had split them to make one figure for each site in the previously revised manuscript, following the editor's comment to ensure that the color schemes used allow readers with color vision deficiencies to correctly interpret our findings. Therefore, we would prefer to keep these figures as is. However, it was not necessary to put hourly precipitation data in all the figures, so we have deleted those from Figs. 4, 5(i)-(p), and 6(i)-(p) in the revised manuscript.

*1-6) There are no x-axis titles in Fig. 3, 4, 5, 6, 10, 11, 12, 13, 14, 15.*

→ Following the reviewer's comment, the authors have added x-axis titles of the form "Year/Month" in Figs. 3, 4, 5, 6, 10, 11, 12, 13, 14, 15, 16 of the revised manuscript.

*1-7) The resolution of Figure 7 is low. It is difficult to identify the yellow character inside the figure.*

→ Thank you for the comment. The figure has been revised appropriately. Also, one figure was not necessary, so it has been removed.

*1-8) Table 1 : Please add the html addresses for GEBCO, Japan Meteorological Agency website, Ministry of the environment website, respectively.*

➔ The html addresses of the three references have been added.

*1-9) Table 3 : Are the days when omega in Hinase and Shizugawa in the simulation with RCP8.5 opposite (365 and 216)? Please check it.*

➔ Thank you for the comment. The authors have recalculated and found the number was wrong although the model results shown in Figs. 14 and 15 were correct. Therefore, we have corrected the values in Table 3 and relevant sentences in the main text (Lines 407-420 in the revised manuscript). However, the modification does not affect the purport of this study.

*1-10)The caption about DIC has been forgotten in Figure 5.*

➔ Thank you for the comment. The caption about DIC has been added to Fig. 5 in the revised manuscript.

**Chapter 1**

*1-11) In Chapter1, author explains the mechanisms how global warming occurs and ocean uptake carbon in the second and third paragraphs. I recommend you add the equation such as (CO2 + H2O -> H2CO3 etc) after the appropriate sentences (2nd and 4ᵗʰ paragraphs).*

➔ Based on the comment, the following equation has been added to the revised manuscript (Line 49 in the revised manuscript):
$$CO_2 + H_2O \rightarrow H_2CO_3 \rightarrow HCO_3^- + H^+. \qquad (1)$$

*1-12) 4th paragraph: Is the word "CaCO3 saturation state (Ω) values" is general word?*

➔ The term "values" is not necessary, so has been deleted (Line 65 in the revised manuscript).

*1-13) 5th and 10th paragraph: "① However, it is not clear when and where these effects occur in the ocean. Therefore to assess the acidification impact on commercially …… and evaluate the impacts on each species" "② Although the ecological effects of coastal warming, acidification…..clearer, when and how these effects will occur at oyster-farming sites are unknown." These are duplicated. I think you can delete one of them (①).*

➔ The authors have deleted ① in the revised manuscript. Thank you for the suggestion.

**Section 2.1**

*1-14) The 4th paragraph can be deleted, judging from the contents and balance of the manuscript.*

➔ We have deleted the paragraph following the reviewer's comment.

*1-15) 5th paragraph: If you revise the figure, I recommend you to add the location name "Chikusa River", "Katakami Bay", "Genju bay", and "Hachiman River", respectively.*

→ The location name has been added to Fig. 1 in the revised manuscript.

**Section 2.4**

*1-16) 2nd paragraph: The model domain area can be shown in Figure 1 with observational sites.*

→ The model domain area has been shown in Fig. 1(b), (d) in the revised manuscript.

*1-17) 3rd paragraph: Please add number in km for "15 arc-second".*

→ 15 arc-second corresponds to around 500 m, so this information has been added to Line 216 in the revised manuscript.

**Section 2.4.1**

*1-18) Section 2.4.1 can be combined into Section 2.4, which results in deleting Section 2.4.1.*

→ Following the reviewer's comment the paragraph has been merged into Section 2.4 and Section 2.4.1 has been deleted (in Lines 220-225 in the revised manuscript).

**Section 2.5 and Table 2, 3, and 16**

*1-19) I couldn't understand how the end date of the spawning season of pacific oysters with RCP 8.5 senario is determined, judging from your sentence in Section 2.5 "Pacific oysters reach sexual maturity when the accumulated water temperature reaches 600 degree based on a water temperature of 10 degree, and that at water temperatures of 20 degree or higher they spawn ocean and then mature and spawn again". Is it possible to find the end of the spawning day, although we cannot find the start day with RCP 8.5 senario in Sizugawa?*

→ The previous manuscript did not clearly describe the end of spawning assumed in this study. Therefore, the authors have added the following description in the revised manuscript (in Lines 231-233 in the revised manuscript): *"In this study, based on Oizumi et al. (1971), spawning was assumed to start when the accumulated water temperature reaches 600 (℃ day) based on a water temperature of 10 (℃) and to end when the water temperature drops below 20 (℃)."*

**Section 3.1**

*1-20) Please correct typo "oin" in the 1st paragraph in Section 3.1.*

→ Corrected ("*on August*") (in Line 247 in the revised manuscript). Thank you for the comment.

*1-21) 2nd paragraph can be combined into the 1st paragraph.*

→ Combined.

*1-22) Please add figure numbers after your sentence, which makes more readers friendly.*

➔ Following the comments, the authors have added figure numbers after sentences in the entire text.

*1-23) 4th paragraphs: you write "a statistically significante relation between the rainfall and salinity was not idfentified" and "the relation between the salinity and rainfall was not statistically significant at any of the sites in Hinase and Shizugawa, and future studies are necessary". I think that the reason why you cannot find the significant relationship is because you compared the total records when you compared. Please try to analyze with some ingenuities. For example, comparison of the short-term data, individually. When we see the time-series of salinity and precession visually, we can see that they seemingly have some relationships.*

➔ Based on the reviewer's helpful comment, the authors have checked the salinity and precipitation data again. As we have already found and the reviewer also pointed out, there was no significant relation found statistically. However, if we focused during the low-salinity and rainfall events, we found that rainfall does not always result in a significant decrease in salinity, but when there is a significant decrease in salinity, it always tends to be after rainfall events. Therefore, the authors have modified this paragraph by adding the information and rephrasing sentences in the previous manuscript (in Lines 270-274 in the revised manuscript).

*1-24) 5th paragraph: Is there no figure we should see? If so, please add the sentence "not shown".*

➔ Added (ine Line 276 in the revised manuscript).

**Section 3.2**

*1-25) 1st paragraph: Please add some description about the horizontal distributions more.*

➔ Some description has been added as follows (Lines 314-321 in the revised manuscript): "*The model successfully reproduced the spatio-temporal variations of each parameter in Hinase and Shizugawa (Figs. 8 and 9), such as significant seasonal fluctuation of water temperature (Figs. 8(a)-(d), 9(a)-(d)). The modeled salinity was relatively uniform in space and season. However, the salinity was lower in coastal regions in Hinase, especially near rivers in summer where and when freshwater discharge from rivers are dominant (Fig. 8(e)-(h)). The spatio-temporal variability was less in Shizugawa, although the seawater flowing into the bay is likely influenced by freshwater discharged from the Kitakami River, the fifth longest river in Japan (Fig. 9(e)-(h)). The modeled DO is in direct contract with water temperature, higher in winter and lower in summer (Figs. 8(i)-(l) and 9(i)-(l)), primarily caused by higher and lower solubility of oxygen in cooler and warmer water, respectively.*"

**Section 4.2**

*1-26) 2nd sentence "Our model results imply that the number of ... in Hinase will increase from .. to 365 days with the RCP8.5 scenario in the 2090s". Is this sentence correct? Please check it.*

→ As mentioned above (1-9), the authors have recalculated and found that the values shown in the previous manuscript were incorrect. Therefore, we have corrected the numbers in Table 3 and in this sentence (Lines 407-412 in the revised manuscript). Again, the modification does not affect the purpose of this study and figures.

**Section 5.1 and 5.2**

*1-27) These sections are written about the mitigation and adaption. I think that it may not include them in the scientific paper. I recommend you delete these sections.*

→ These sections have deleted following the reviewer's comment.

[Figure]

**Responses to Reviewer #2's comments**

Thank you for all your comments that greatly helped us in improving our manuscript. We have revised and re-submitted our manuscript based upon all your comments. Most of the previous figures have also been revised following your and another reviewer's comments. The information about an English certificate in the previous manuscript has also been deleted in the revised manuscript.

*(Line 170) The calculation of pH using CO2SYS requires the total boron concentration, bisulfate dissociation constant, and hydrogen fluoride dissociation constant in addition to the acid dissociation constant of Lueker et al. [2000] used by the authors. Please describe the values used here.*

→ Thank you for the helpful comment. The authors have modified the relevant sentence by adding the necessary information as follows (in Lines 161-164 in the revised manuscript): *"The pH (total scale) values at the in situ temperatures were calculated from the carbonate dissociation constants in Lueker et al. (2000), the total boron concentration in Lee et al. (2010), the bisulfate dissociation constant in Dickson (1990), and the hydrogen fluoride dissociation constant in Perez and Fraga (1987), and temperature, salinity, TA, and DIC using CO2SYS (Pierrot et al., 2006)."*

*(Figure 8 and 9) It is recommended to include isopleths in the figures, such as at intervals of 2 °C for water temperature. This inclusion can facilitate comprehension in individuals with color vision deficiencies.*

→ Thank you for the comment. The authors have modified Figs. 8 and 9 accordingly.

*(Figure 14 and 15) Modifying the range of the vertical axis would enhance the comprehension of seasonal variations and the distinctions between current and future projections. For instance, a range of 25-35 for salinity and 2000-2400 µmol kg-1 for TA would provide an adequate representation.*
*It is recommended to overlay Figure 14(g) and 15(g) with the respective current and predicted spawning periods shown in Figure 16 under each scenario. This visualization would facilitate readers' comprehension of the overlap between the extended spawning period and the period of low aragonite saturation. Additionally, the number of figures is too large and can be reduced by this aggregation.*

→ Former Figs. 14 and 15 have been modified accordingly and former Fig. 16 has been deleted, based on the comment. However, to respond to another reviewer's major comment, the authors have added one figure (Fig. 14 in the revised manuscript). Therefore, former Figs. 14 and 15 have been renumbered as Figs. 15 and 16, respectively, and therefore, the total number of figures remained the same.

---

## Author Response (AR3)

**Responses to Reviewer #1's comments**

First of all, the authors would thank the reviewer for giving practical comments. We have revised our manuscript based upon all your comments. Below are our responses to the comments:

**Response to major comments**

1) *I am suggesting major revisions largely due to the discrepancies between the modeled and observed results. I think there needs to be more discussion as to why the model doesn't pick up the short term variability, and how that impacts your conclusions about when thresholds will be met. The models seem to do a good job predicting average seasonal variability, but there is a lot of variation over shorter timescales that may push oysters to experience harmful pH and oxygen conditions over shorter timescales. In the discussion, I think you need to focus less on the modeled outcomes, and more on the observations. What are the conditions now, and what timescales do they vary over? How will that impact oysters? Please revise the discussion to focus on your observations instead of the model output.*

➔ Thank you for the practical comments. The current structure of Section 4 is a result of addressing another reviewer's previous comment to move future projection results from Section 3 to Section 4. Therefore, the reviewer may have felt that this section focuses more on model results rather than the observations. To take the present comments into account, the authors would like to refer the reviewer to Section 3, in which observed results are already described in detail. Moreover, in 4.2, we have also referred to observation-based estimated start and end dates of Pacific oyster spawning periods before discussing the projected start and end dates for the future (in Lines 396-398 in the revised manuscript).

   As the reviewer pointed out, the model did not reproduce well the observed short-term fluctuations in biogeochemical parameters, which may affect the accuracy of future projection results. This was mainly because the temporal resolution of the model output is 6 hours, insufficient to resolve significant short-term fluctuations in biogeochemical processes predominantly caused by biological activities, i.e., photosynthesis by phytoplankton, eelgrass, and seaweeds during the day and respiration of marine organisms at night. Although the spatial resolution of the model (2 km) is relatively high for downscaling climate model outputs, it is insufficient to reproduce spatial differences in biogeochemical-parameter values among the four monitoring sites in Hinase and Shizugawa. Also, the model-observations mismatch for TA and DIC values, especially the failure to reproduce sudden decreases, likely resulted from insufficient input of freshwater from rainfall and riverine water into the model. These have been described in Lines 331-338 in the revised manuscript.

   Considering the reviewer's comment, i.e., to focus more on our observations rather than our model outputs, the authors have added the observed short-term fluctuations of $\Omega_{arag}$ to modeled $\Omega_{arag}$ (Figure S1 below and in the revised manuscript). As a result, the simulated or projected number of days on which $\Omega_{arag}$ values are below the threshold of acidification for Pacific oyster larvae (1.5) has been modified from 0 days to 3 days for the present, from 0 days to 5 days with the RCP 2.6 scenario, and from 204 days to 256 days with the RCP 8.5 scenario in Hinase. In Shizugawa, the simulated or projected number of days have been modified from 0 days to 7 days for the present and with the RCP 2.6 scenario, and from 244 days to 322 days with the RCP 8.5 scenario (Table 3). The results show that consideration of the short-term variations may push oysters to experience harmful pH and $\Omega_{arag}$ conditions, as the reviewer pointed out. Unfortunately, we could not estimate the modified number of days on which DO concentrations are below 203 $\mu$mol kg$^{-1}$ by considering the observed short-term

fluctuation of DO concentration due to the lack of continuous DO observations in this study. As previous reviewers commented that our manuscript has relatively many figures already, we did not add a new figure to the main text but instead have added one as supplementary material (Fig. S1) in the revised manuscript, and have modified Table 3 to reflect what was mentioned above.

Making the above-mentioned modifications following the reviewer's comments have also stressed to the authors the need to describe more clearly the discrepancy between the scientific findings and the fact that no specific impacts of ocean acidification on Pacific oyster larvae have so far been detected in the study sites, even they occasionally experience the critical level of ocean acidification proposed by a previous study ($\Omega_{arag} < 1.5$; Waldbusser et al., 2015). The relevant descriptions have also been added to Lines 435-439 in the revised manuscript.

These modifications do not change the overall purpose of this study. However, the following description, which was part of the Abstract in the previous manuscript, has been deleted from the revised manuscript, because if we take the short-term fluctuation in DO concentration into account, the description is not appropriate:

*"On the other hand, no significant impact of surface-water deoxygenation on Pacific oysters was identified at present nor was projected for the future in both sites."*

[Figure]

Figure S1. Simulated or projected $\Omega_{arag}$ in Hinase from August to July (left) and in Shizugawa from September to August (right) for the present (top), for the 2090s with the RCP 2.6 scenario (middle), and for 2090s with the RCP 8.5 scenario (bottom). Solid black lines, dashed blue lines, and dotted brown lines are identical to results shown in Figures 15(g) and 16(g). Solid green lines denote modeled daily $\Omega_{arag}$ minima if present day observed daily $\Omega_{arag}$ fluctuations are included.

**Responses to specific comments**

2) *L 47 and 50: I don't think 'leached' is the correct word here. I would say CO2 is absorbed by the oceans from the atmosphere.*
➔ The term "leached into" has been replaced with "absorbed by" (in Lines 45 and 48 in the revised manuscript).

3) *L127 (and throughout): Please refer to 'Chapters' as 'sections' instead.*
➔ The terms "Chapters" have been replaced with "sections" in the entire text (in Lines 108 and 110 in the revised manuscript).

4) *L179: Should 'alkaline' be 'alkalinity' here?*
➔ Thank you for the comment. That should be "alkalinity" and has been modified accordingly (in Line 156 in the revised manuscript).

5) *L206: You correct for the drift of the sensor, not the 'observed value'*
→ The reviewer is right, and we have modified the text accordingly (in Lines 163-164 in the revised manuscript).

6): *L248: 'biochemical' should be 'biogeochemical'; check manuscript for other instances*
→ The term "biogeochemical" has been replaced with "biogeochemical" (in Line 207 in the revised manuscript). We have also checked the entire manuscript and have confirmed that there are no other instances.

7) *L333: replace 'vicinal' with 'nearby'*
→ The term "vicinal" has been replaced with "nearby" (in Line 272 in the revised manuscript).

8) *L413: replace 'creatures' with 'organisms'*
→ The term "creatures" has been replaced with "organisms" (in Line 334 in the revised manuscript).

9) *L439/443: replace 'contributed to by' with 'controlled by'*
→ The term "contributed to by" has been replaced with "controlled by" (in Lines 360 and 364 in the revised manuscript).

10) *L548: Change section name to 'Conclusions'*
→ We have changed the section name to "Conclusions" (in Line 457 in the revised manuscript).

---

## Author Response (AR4)

**Responses to Reviewer #1's comments**

The authors would like to thank the reviewer for the comments. We have revised our manuscript based upon all your comments. Below are our responses:

Response to minor comments:

*P9, l252-l253, I suggest you this revision below.*
*"In Hinase, Pacific oysters were estimated to have stopped spawning between October 24 and November 4, 2020, and between October 25 and November 7, 2021 and to have begun spawning between June 8 and19 in 2021, judging from the water temperature thresholds based on Oizumi et al. (1971)".?*
→ We have added the phrase "judging from the water temperature thresholds based on Oizumi et al. (1971)" in the revised manuscript (in Line 257).

*P9, l252-253, Please add the Figure number we should see.*
→ There is no specific figure to see, but instead, the authors have added the table number (Table 2) to see (in Line 255 in the revised manuscript).

*P12 eq.(4) please check the location of the comma(,).*
→ We have confirmed the location of the comma is appropriate and is not subscript, either.

*P12, l345-l359, Fig14, please write which location you used when you draw Fig. 14.*
→ Thank you for the comment. We have added the information of location we used in the caption for Fig. 14 (in Hinase (at H-4) and in Shizugawa (at S-4), respectively) (in Line 891 in the revised manuscript).

*P12, l361-l362, I wonder if you need the consideration from the effect of air-sea exchange as well as biological production?*
→ Based on the reviewer's comment, the authors checked previous papers that use Equations (3) though (6) (e.g. Hauri et al., 2013), but could not find any description of the effect of air-sea exchange on the term of $\partial pH/\partial DIC * \Delta DIC$. We suppose the effect is rather reflected in the term of $\partial pH/\partial T * \Delta T$ as a result of the change in the solubility of $CO_2$ due to a change in temperature. Therefore, we have retained the description here.

[Figure]

**Responses to Reviewer #2's comments**

The authors would like to thank the reviewer for the comments. We have revised our manuscript based upon all your comments. Below are our responses:

Response to comments:

*There are frequent mentions to a longer spawning period leading to a shorter shipping period, but the relationship between these two is not clear. I kindly request an explanation of this relationship to be added where it is first mentioned in the text (around line 230).*
→ To clarify the relation, the sentence has been modified as: "Therefore, there is a concern that a rise in water temperatures in the future may cause earlier or longer spawning and maturation times. The earlier spawning and maturation times may result in a mismatch with existing oyster-farming approaches. The prolonged spawning period may shorten the oyster shipping period and lower their quality (Akashige and Fushimi, 1992), potentially damaging the oyster-processing industry." (in Lines 229-232 in the revised manuscript).

*While you cite previous reports on oxygen thresholds, please consider mentioning the possibility that these thresholds could be higher in the future due to increased acidification; I recommend referring to \*Steckbauer et al. (2020).*
*\*https://onlinelibrary.wiley.com/doi/10.1111/gcb.15252*
→ Thank you for the useful comment. Based on the comment, the authors have added the following sentences in Section 4.2 (in Lines 433-436) in the revised manuscript:
"Also, previous studies imply that impacts on marine organisms appear more severely under the co-occurrence of ocean acidification and deoxygenation than the occurrence of each stressor alone (Steckbauer et al., 2020; Yorifuji et al., 2023). In other words, the threshold for deoxygenation alone (203 $\mu$mol kg$^{-1}$) could be higher in the future when ocean acidification progresses."